# LLM-Augmented Chemical Synthesis and Design Decision Programs

**Haorui Wang** [1]   **Jeff Guo** [2 3]   **Lingkai Kong** [4]   **Rampi Ramprasad** [1]
**Philippe Schwaller** [2 3]   **Yuanqi Du** [† 5]   **Chao Zhang** [† 1]

## Abstract

Retrosynthesis, the process of breaking down a target molecule into simpler precursors through a series of valid reactions, stands at the core of organic chemistry and drug development. Although recent machine learning (ML) research has advanced single-step retrosynthetic modeling and subsequent route searches, these solutions remain restricted by the extensive combinatorial space of possible pathways. Concurrently, large language models (LLMs) have exhibited remarkable chemical knowledge, hinting at their potential to tackle complex decision-making tasks in chemistry. In this work, we explore whether LLMs can successfully navigate the highly constrained, multi-step retrosynthesis planning problem. We introduce an efficient scheme for encoding reaction pathways and present a new route-level search strategy, moving beyond the conventional step-by-step reactant prediction. Through comprehensive evaluations, we show that our LLM-augmented approach excels at retrosynthesis planning and extends naturally to the broader challenge of synthesizable molecular design.

## 1. Introduction

Retrosynthesis (Corey & Wipke, 1969; Corey et al., 1985) concerns with breaking down a target molecular structure into a sequence of simpler or more readily available precursor structures and chemical reactions (Boström et al., 2018). It is essential for many chemistry problems that require the realization of proposed molecular structures from organic synthesis to drug discovery (Blakemore et al., 2018). Nevertheless, the search space for a given target is tremendous as the number of possible synthesis pathways grows exponentially with the number of reaction steps or the depth of the route tree. Consequently, efficient decision-making in retrosynthesis planning, and more broadly, in chemical design, remains a critical challenge.

Recent research has harnessed machine learning to tackle retrosynthesis by modeling reactions with a single-step model which predicts a reaction template, i.e., a reaction coded as a pattern, to synthesize the given target molecule (Segler & Waller, 2017; Coley et al., 2017; Liu et al., 2017), including graph neural networks (Dai et al., 2019; Chen & Jung, 2021), and subsequently reverse the template to obtain the reactants. Another branch of single-step models do not rely on the provided reaction templates and directly predict reactants (Liu et al., 2017; Schwaller et al., 2020; Igashov et al.). After training the single-step models, they are further connected with a search algorithm (e.g. Monte Carlo tree search (Segler et al., 2018) or A* search (Chen et al., 2020) to perform multi-step retrosynthetic analysis, which halts when a path to a set of predefined purchasable molecules is found.

Recent studies have shown that large language models (LLMs) implicitly encode substantial chemical knowledge, as evidenced by their remarkable performance in searching molecular structures with optimized properties (Wang et al., 2024). In addition, LLMs have been leveraged for reasoning and planning problems, such as automated experimentation in chemistry (M. Bran et al., 2024; Boiko et al., 2023). Despite the apparent promise, the extent to which LLMs can handle tightly constrained decision processes, such as retrosynthesis planning, remains largely unexplored. Unlike open-ended tasks like text generation, retrosynthesis imposes rigorous constraints on the sequence of actions (reaction steps). Only certain reaction templates are valid, and only commercially available or otherwise feasible precursors can be used.

In this paper, we investigate whether the knowledge embedded in LLMs can be effectively leveraged for complex sequential decision-making tasks in chemistry such as retrosynthesis planning. Crucially and by contrast to existing LLM works for retrosynthesis (Nguyen-Van et al., 2024; Yang et al., 2024), we *do not* tune the base LLM. Instead, by exploring how LLMs perform under heavy constraints,

---

[*]Equal contribution   [1]Georgia Tech   [2]École Polytechnique Fédérale de Lausanne (EPFL)   [3]National Centre of Competence in Research (NCCR) Catalysis   [4]Harvard University   [5]Cornell University. Correspondence to: Haorui Wang <hwang984@gatech.edu>.

*Proceedings of the $42^{nd}$ International Conference on Machine Learning*, Vancouver, Canada. PMLR 267, 2025. Copyright 2025 by the author(s).

we aim to gain insights into their potential to serve as powerful decision-making engines, ultimately advancing our understanding of their capabilities in chemistry and beyond. Furthermore, we expand the scope to study the capability of LLMs in not only finding a synthesis pathway, but also simultaneously optimizing the property of the target molecule, known as synthesizable molecular design (Bradshaw et al., 2019; 2020; Gottipati et al., 2020; Horwood & Noutahi, 2020; Korovina et al., 2020; Gao et al., 2022; 2024; Koziarski et al., 2024; Cretu et al., 2024; Seo et al., 2024; Swanson et al., 2024). We summarize our main contributions are as follows:

- ▷ We propose an efficient and effective way to encode the sequence of synthesis decisions: (1) a language to describe reactions that LLMs understand and (2) efficient data structures to store the exponential-growth tree-structured synthesis pathways.

- ▷ We integrate a sequence-level search strategy into LLM retrosynthesis planning, sampling complete decision sequences (full multi-step pathways) instead of single reaction steps, and apply a smooth reward with partial feedback to evaluate each pathway.

- ▷ Experimentally, we study both the retrosynthesis planning and synthesizable molecular design problems in this unifying paradigm of LLM-augmented reaction decision program.

## 2. Problem Formulation

We formulate the retrosynthesis planning problem as a sequential decision-making problem. At the core of this task is a **molecule set**, which contains either the molecules we aim to synthesize (target) or purchase directly (permitted commercial building blocks). We initialize the molecule set with only the **target molecule** and evolve over successive search steps until there is no molecule in the set that is non-purchasable.

At each step, we use a **backward reaction** to decompose a molecule in the set (the product) into its reactants. This involves removing the product from the molecule set and adding the corresponding reactants generated by the selected backward reaction. The process terminates when either all molecules remaining in the set are purchasable or the maximum budget of attempts is reached.

A reaction is formally defined by a **reaction template**, which specifies a structural transformation pattern in the form of a SMARTS string (Daylight Chemical Information Systems). We denote the set of feasible reaction templates by $\mathbb{T}$ and the set of purchasable compounds by $\mathbb{C}$. Both $\mathbb{T}$ and $\mathbb{C}$ are flexible and can be refined or expanded

without altering the underlying framework, ensuring adaptability to various chemical spaces.

Given $\mathbb{T}$ and $\mathbb{C}$, our goal is to iteratively select backward reactions that construct a valid synthetic route for the target molecule. Each molecule in the synthetic route, including intermediates, is explicitly defined through the application of reaction templates to its reactants. Compared to general molecule generation tasks, retrosynthesis planning introduces additional challenges, such as enforcing chemical reaction rules and ensuring the use of commercially available building blocks.

### 2.1. Retrosynthesis Planning

Given a target molecule $M_{\text{target}}$, the objective is to identify a sequence of reactions $\{r_1, r_2, \ldots, r_n\}$ such that:

1. $M_{\text{target}}$ can be recursively decomposed into reactants by applying reaction templates from $\mathbb{T}$.

2. The final set of reactants consists exclusively of molecules in $\mathbb{C}$.

3. Each reaction $r_i \in \mathbb{T}$ is chemically valid and adheres to the predefined reaction rules.

At each decision step $t$, we select a reaction $r_t \in \mathbb{T}$ to apply to a molecule $M_t$ in the molecule set. This generates its reactants $\{M_{t,1}, M_{t,2}, \ldots\}$, which are then added to the molecule set, replacing $M_t$. The task is completed when all terminal nodes in the synthetic pathway correspond to molecules in $\mathbb{C}$.

### 2.2. Synthesizable Molecular Design

In contrast to retrosynthesis planning, we consider the **synthesizable molecular design** problem, where the goal is to find molecules with optimal properties evaluated by an oracle function $O$, while simultaneously ensuring that they are synthetically accessible through feasible reaction pathways.

$$\arg\max_{m \in \Omega} O(m) \ \text{ s.t. } V(R(m)) = 1$$

where $\Omega$ is the set of generated molecules, $R(\cdot)$ returns the synthesis path, and $V(\cdot)$ checks the validity of the path.

## 3. Methodology

### 3.1. Route formatting

Traditional machine learning methods directly predict reaction classes or reactants based on input molecules, which by definition, defines the synthesis route when the full set of reaction classes and molecules are considered (Zhong et al., 2024). However, using LLMs as retrosynthesis route generators necessitates a well-defined textual input-output format,

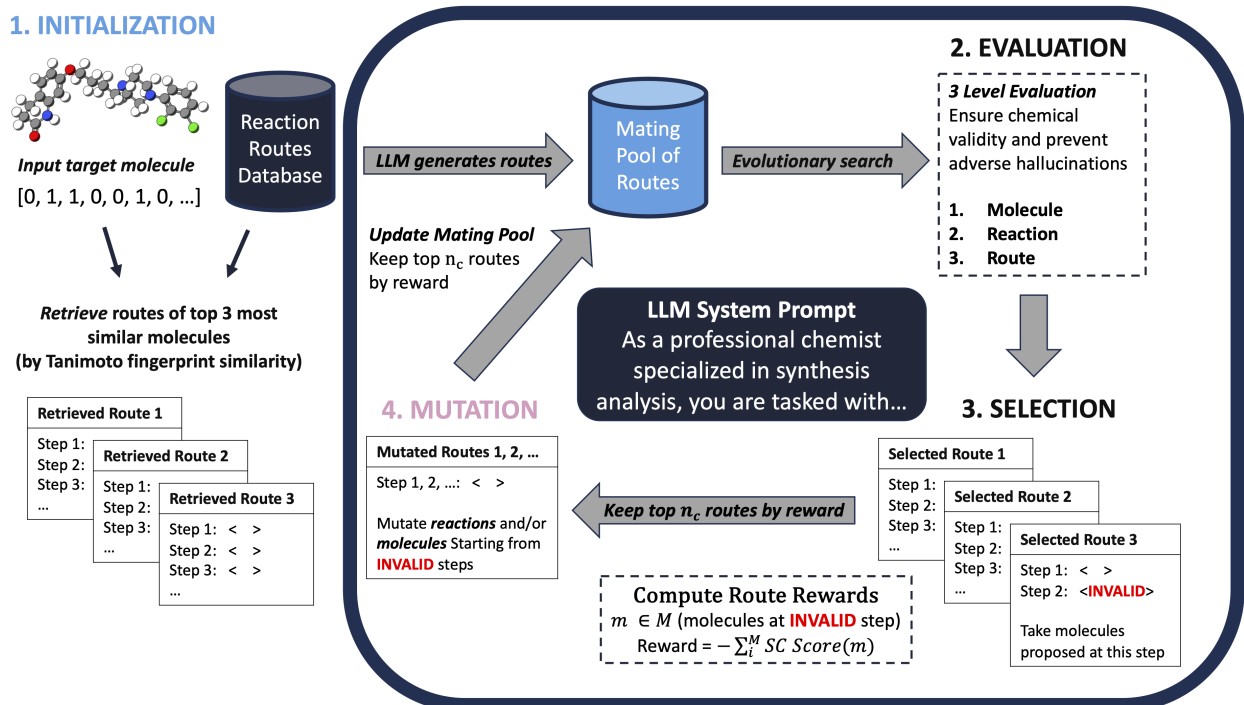

Figure 1. Overview of the LLM-Syn-Planner. **1. INITIALIZATION:** Based on the target molecule, reaction routes of similar molecules are retrieved and scored by the SC score (Coley et al., 2018). **2. EVALUATION:** The LLM generates new routes which are evaluated. **3. SELECTION:** Starting from invalid steps in the reaction routes, the SC score of the molecules at this step are computed and the top $n_c$ routes are selected. **4. MUTATION:** Starting from these invalid steps, the LLM proposes mutations to modify the molecules and/or reactions at this step. Repeat until a solution is found or the budget is reached.

as LLMs are highly sensitive to prompt design (Sclar et al., 2024). A critical challenge lies in determining how to represent the retrosynthesis route for LLMs. Prior research have proposed two main representation formats:

▷ Textual descriptions (Liu et al., 2024a): Textual descriptions align naturally with the text generation capabilities of LLMs and uses descriptive language to detail each reaction step. However, the flexibility and lack of standardization in textual descriptions make it challenging to consistently extract essential information, such as reactants, products, and reactions. This ambiguity complicates the evaluation of individual steps and the validation of the overall synthesis route.

▷ Tree structures (Chang et al.): Tree structures (Figure 2a) represent synthetic pathways as hierarchical trees, capturing the relationships between reactants and products in a structured manner. While tree structures provide a more systematic representation, their complexity increases significantly in multi-step retrosynthesis tasks, leading to deeply nested structures that can overwhelm the LLM's reasoning capabilities.

To address these limitations, we draw inspiration from tradi-

tional tree search-based approaches to retrosynthesis planning (Segler et al., 2018). In these approaches, the nodes in the search tree represent synthetic states, and the tree itself contains all molecules required to synthesize the target molecule at the root. A target molecule is considered synthesized when all leaf nodes in the tree correspond to purchasable building blocks. The edges of the tree correspond to reactions, which specify a chemical transformation between states of connected nodes.

Building on this framework, we reformulate retrosynthesis planning into a step-by-step decision-making process that is more suitable for LLMs (Figure 2b). Specifically, we represent the synthesis route as a sequence of decisions, where each step involves proposing a reaction from a database of reaction results, i.e., *reaction templates*. The LLM maintains a dynamic molecule set that starts with the target molecule and evolves as reactions are selected, ending when all molecules in the set are purchasable. To improve the decision-making process, we integrate a reasoning component called the "Rational" at each step. This reasoning step encourages the LLM to think before making decisions (Wei et al., 2022). Additionally, we ask the LLM to explicitly output the product and reactants in each step, in order to keep the generated

route more consistent.

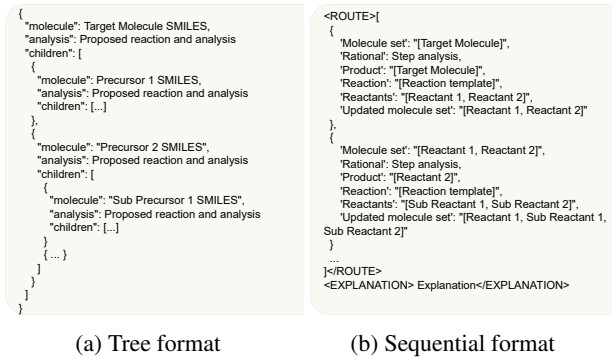

(a) Tree format      (b) Sequential format

*Figure 2.* Different route formats for retrosynthesis planning

## 3.2. LLM as a single-step prediction model

Recent studies have demonstrated the potential of utilizing LLMs as planners for complex decision-making tasks (Song et al., 2023; Huang et al., 2024). A common approach is integrating LLMs with traditional search algorithms such as MCTS (Zhao et al., 2024) and A* search (Zhuang et al., 2023). This integration addresses a key limitation of LLMs: their lack of a systematic mechanism to explore structured solution spaces. Without such mechanisms, LLMs may struggle to effectively navigate complex decision-making scenarios. The core idea of these methods is straightforward: treat the LLM as a policy that directly generates the next action based on the history of past actions and observations. Meanwhile, search algorithms like MCTS and A* systematically explore and optimize the solution space, ensuring robustness and completeness.

Building on this, we propose using LLMs as single-step retrosynthesis predictors, and operate in the template-based approach, where we start with a pre-defined templates set that represents all the reactions the LLM *could* suggest and a reference reactions database based on USPTO. The product molecule in each step serves as the input, and the LLM is queried to predict a reaction that synthesizes this product molecule. To do this, we first task the LLM with identifying substructures and functional groups in the product molecule. Next, we draw inspiration from Coley et al. (2017) and compute the Tanimoto similarity between the substructures and the product molecules in the reference reactions database. The hypothesis is that similar product molecules are synthesized from similar reactions. Following these steps, the LLM *retrieves* a template from the pre-defined list, which is important as it removes any possibility of hallucinated templates. The template is then applied to the product molecule to obtain a set of predicted reactants.

By contrast to existing single-step prediction models (Maziarz et al., 2023), it is non-trivial to obtain a proba-

bility of choosing a template from an LLM. Therefore, we assign pseudo-probabilities to the predicted reactions by employing *self-consistency frequency*, which is an ensemble approach that samples $k$ independent reactions for the next step, denoted as $\{r_{t+1}^{(j)}\}_{j=1}^{k} \sim p(r_{t+1}|m)$ at step $t$. From these samples, we identify the unique reactions and consider them as the set of potential next-step reactions. The frequency of each reaction in this set is then used to compute its cumulative score, given by:

$$p(n) = \frac{\#\{j \mid r_{t+1}^{(j)} = n\}}{k},$$

where $\#\{j \mid r_t^{(j)} = n\}$ denotes the count of samples for reaction $n$. In essence, this expression computes how many times each reaction appears in $k$ sampled reactions.

Finally, we integrate the LLM as a single-step predictor with MCTS (Segler et al., 2018) or Retro* (Chen et al., 2020) search algorithms to explore retrosynthesis pathways.

---

**Algorithm 1** LLM-Syn-Planner Algorithm

---

**Data:** The target molecule $T$; the reward function $F$; the evaluation function $E$; the population size $n_c$; the number of retrieval size $n_o$; the routes retrieval set $\mathbb{O}$; the maximum number of attempts `budget`.

**Result:** Found synthesis routes population $\mathbb{P}^*$

**begin**

  $\mathbb{P}_0$ = []

  **while** *len($\mathbb{P}_0$) < $n_c$* **do**

    sample $\mathbb{P}_o = \{p_i\}_{n=1}^{n_o}$ from $\mathbb{O}$ proportionally to their products' Tanimoto similarity to $T$

    $\mathbb{P}_0$.append(INITIALIZATION($T, \mathbb{P}_o$))

  **for** $p \in \mathbb{P}_0$ **do**

    Compute $F(p)$

  **for** $t \in [0, \texttt{budget}]$ **do**

    offspring = []

    **for** *num_mutations* **do**

      sample $p$ from $\mathbb{P}_t$ proportionally to reward $F(p)$

      evaluate $p$ using the evaluation function $E(p)$ to get feedback $f$

      offspring.append(MUTATION($T, p, f$))

    **for** $p \in \mathbb{P}_t$ **do**

      Compute $F(p)$

    $\mathbb{P}_{t+1} \leftarrow$ sorted($\mathbb{P}_t$)[: $n_c$]

  Return $\mathbb{P}_{\texttt{budget}}$

---

## 3.3. LLM as a synthesis pathway sampler

Although LLMs can leverage search algorithms to explore the search space, akin to existing works that pair single-step reaction prediction with search algorithms (Zhong et al., 2024), we are particularly interested in their ability to design synthesis routes directly for a given target molecule.

| Level | Type | Explanation |
|---|---|---|
| Molecule | Validity | Whether the molecule is valid (RDKit parsable) |
| | Availability | Whether the molecule is commercially available, i.e., in the building block stock |
| Reaction | Existence | Whether the reaction exists in the database |
| | Validity | Whether the product can be synthesized from the proposed reactants |
| Route | Connectivity | Whether the route is connected |

*Table 1.* Three levels of feedback in the evaluation stage.

To this end, we propose an evolutionary search algorithm named LLM-Syn-Planner that enables LLMs to *generate* and optimize the whole retrosynthetic pathways directly. We emphasize *generate* as the LLM is not explicitly retrieving a reaction template like in the case of using the LLM as a single-step prediction model and coupling a search algorithm. Unlike existing works that follow this paradigm (Zhong et al., 2024), our approach generates the entire multistep synthesis tree directly. The algorithm operates as follows: Given a target molecule, we first generate an initial pool of retrosynthetic routes using INITIALIZATION queries from LLMs, where each route is evaluated using a reward function, $F(\cdot)$. Next, a route is sampled with a probability proportional to its reward and edited using a MUTATION operator to generate offspring. This mutation process is repeated *num_mutation* times, after which the newly generated offspring are added to the population. The offspring are then evaluated using $F(\cdot)$, and the $n_c$ fittest candidates at each step are selected to pass on to the next generation. This iterative process continues until the maximum number of model calls is reached. The overall workflow consists of four key stages: (1) **Initialization**, (2) **Evaluation**, (3) **Selection**, and (4) **Mutation**. This process is outlined in Algorithm 1.

**Initialization.** In the INITIALIZATION function, we query the LLM to generate initial retrosynthesis routes for the target molecule. To enhance its predictions, we employ a molecular similarity-based retrieval-augmented generation (RAG) approach, providing reference routes for the LLM. Specifically, we use the Morgan molecular fingerprint with Tanimoto similarity to identify structurally similar molecules in a database and retrieve their corresponding synthesis routes. We then provide the synthesis routes of the top three most similar molecules as references to the LLM.

**Evaluation.** We propose a three-level evaluation process to assess the quality of each step in the generated synthetic route: molecule level, reaction level, and route level, as shown in Table 1.

At the molecule level, we validate whether the molecules in the molecule set are both valid and purchasable. At the reaction level, we first perform reaction mapping to verify the reactions proposed by the LLM. This involves grounding and matching them against a reaction database. We begin by searching for exact matches. If no exact match is found, we retrieve the top 100 most similar reactions based on reaction fingerprint similarity. These candidates are then filtered by assessing whether the proposed reaction is chemically feasible for the given product molecule, as even if the retrieved route is for a similar molecule, slight differences in the target molecule structure can render the reaction incompatible. The most similar valid reaction is retained as the matched reaction. Finally, we replace the original reaction proposed by the LLM with the identified match, thus removing the possibility of a hallucinated reaction that we cannot easily verify the chemical soundness of. If no valid match is found in this process, we label the reaction as non-existent. At the route level, we evaluate route connectivity by checking whether the 'molecule set' in a given step aligns with the 'updated molecule set' from the previous step and whether the expected 'product' appears in the current step's molecule set. A step is considered valid if all evaluations pass, except for molecule availability.

**Selection.** The selection stage is the foundation of our evolutionary framework, ensuring the maintenance and progression of a population of candidate routes. In retrosynthesis planning, the success rate is commonly used to evaluate a route's quality. However, within the evolutionary framework, most current routes are unsuccessful. Therefore, we introduce a partial reward mechanism based on SC score (Coley et al., 2018) to assess these incomplete routes. Given a route, we traverse it from the first step sequentially to identify the first invalid step. The molecule set at this step, denoted as $\mathbb{M}$, is then used to compute the reward for the route. The reward function $F(\cdot)$ is defined as follows, where $\mathbb{C}$ is the set of purchasable compounds:

$$F(\mathbb{M}) = - \sum_{m \in \mathbb{M}, m \notin \mathbb{C}} \text{sc\_score}(m)$$

The top $n_c$ routes, as ranked by SC score are selected as the population for the next round of evolution.

**Mutation.** To optimize the current route, we explore the flexibility of LLMs in synthetic route reproduction. Specifically, we enable the LLM to analyze and edit the current route through prompt-based mutation. The LLM is instructed to modify the existing route or propose an alternative if deemed necessary, incorporating evaluation results from multiple perspectives as feedback. If the current route contains reaction-level errors, we retrieve reference routes from $\mathbb{O}$, weighted by their products' Tanimoto similarity to the 'product' molecule in this step and provide them to the LLM. Additionally, for mutation queries, we retain the valid steps of the current route and provide the LLM with only the partial route starting from the first invalid step.

## 3.4. Optimization for synthesizable molecular design

LLM-Syn-Planner can be easily extended to design optimized molecular structures alongside their corresponding synthesis pathways. A simple approach is to first optimize a molecular structure for the desired properties and then determine its synthesis pathway. As a proof of concept, we propose LLM-Syn-Designer, which integrates MolLEO (Wang et al., 2024) as the molecular structure optimizer, which leverages LLMs as genetic operators for molecular optimization. Specifically, we ask the LLM to generate a synthesizable molecule and analyze the synthetic route during the optimization process. To ensure synthesizability, we filter out molecules proposed by LLMs if their SC score exceeds 3.5 at each iteration of the optimization process. Additionally, in every round of evolutionary search, our framework acts as the synthesis pathway finder for the generated molecules. By combining these components, the integrated framework enables the end-to-end design of synthesizable molecules, harnessing the power of LLMs for both molecular optimization and synthesis planning.

## 4. Experiments

### 4.1. Experimental Setup

**Dataset.** We conduct experiments using the USPTO (Schneider et al., 2016; Dai et al., 2019) and Pistachio (pis) datasets. For USPTO, we utilize USPTO-190 (Chen et al., 2020) and a simplified subset, USPTO-EASY, which is randomly sampled from the test set used in Retro* single-step model training. For the Pistachio dataset, we adopt the version from (Yu et al., 2024) but remove the starting material constraints. The route database is constructed using the training and validation sets from Retro*, while the reaction database is a processed version of USPTO-Full, as used in (Yu et al., 2024). For the building block set, we canonicalize all SMILES strings from the 23 million purchasable building blocks available in eMolecules, following the approach of (Chen et al., 2020). We show the statistics of the datasets in Appendix A.1.

**Baseline.** We consider three single-step retrosynthesis models in combination with two search algorithms: MCTS (Segler et al., 2017) and Retro* (Chen et al., 2020). The single-step models are as follows: Graph2Edits (Zhong et al., 2023) is a template-free graph generative model that systematically edits the molecular graph of the target product to generate valid reactant structures. RootAligned (Zhong et al., 2022) is another template-free approach that enforces a strict one-to-one mapping between product and reactant SMILES strings by aligning them to a shared root atom. LocalRetro (Chen & Jung, 2021) is a template-based method that employs local reaction templates involving atom and bond edits, coupled with a global attention mecha-

nism to capture non-local effects.

**Metrics.** For retrosynthesis planning tasks, we use the success rate as the evaluation metric. For synthesizable molecular design tasks, we measure performance using the top-1 expected property in the designed molecules.

**Configuration** [1]**.** We utilize GPT-4o [2] (Hurst et al., 2024) and DeepSeek-V3 (Guo et al., 2025) as our LLMs and set the temperature to 0.7 for all queries, ensuring a balanced trade-off between creativity and reliability. To maintain efficiency, we impose a maximum search time of 60 minutes per molecule.

### 4.2. Retrosynthesis Planning

We present the retrosynthesis planning results in Table 2. The LLM-based approaches show a clear distinction between using LLMs as single-step predictors within a search algorithm and leveraging them to generate complete retrosynthetic routes optimized via tree evolutionary algorithms (LLM-Syn-Planner). When LLMs are integrated into MCTS or Retro*, their solve rates are significantly lower than those of traditional models, particularly on challenging datasets (e.g., Pistachio Hard, where solve rates are near zero). This suggests that current LLM-based single-step models struggle to produce high-quality reaction predictions, leading to suboptimal search performance. Moreover, increasing the number of model calls does not consistently improve results, especially on the USPTO-190 and Pistachio datasets, highlighting intrinsic limitations in LLMs' single-step reaction prediction capabilities.

In contrast, LLM-Syn-Planner performs remarkably well, achieving solve rates comparable to—or even exceeding—some single-step model-guided search. Notably, LLM-Syn-Planner significantly outperforms LLM (MCTS/Retro*), indicating that optimizing full multi-step retrosynthetic routes rather than predicting step-by-step transformations enhances LLM effectiveness. While LLMs may not yet rival expert-designed single-step models in reaction prediction precision, they can generate promising retrosynthetic routes by using their long-term planning capabilities. These findings suggest that the strength of LLMs can be leveraged by reformulating the problem as generating full retrosynthetic pathways that can be optimized through evolutionary techniques. This underscores a potential shift in focus from improving LLMs for single-step retrosynthesis to developing methods that exploit their generative capabilities for full-route planning combined with downstream optimization strategies like EA. We justify the cost of using LLMs in Appendix A.5 and show the case studies

---

[1]Our code is available at https://github.com/zoom-wang112358/LLM-Syn-Planner.

[2]GPT-4o-2024-08-06

| Algorithm | USPTO Easy | | | USPTO-190 | | | Pistachio Reachable | | | Pistachio Hard | | |
|---|---|---|---|---|---|---|---|---|---|---|---|---|
| | Solve Rate (%) | | | Solve Rate (%) | | | Solve Rate (%) | | | Solve Rate (%) | | |
| | N=100 | 300 | 500 | N=100 | 300 | 500 | N=100 | 300 | 500 | N=100 | 300 | 500 |
| Graph2Edits(MCTS) | 90.0 | 93.5 | 96.5 | 42.7 | 54.7 | 63.5 | 77.3 | 88.4 | 94.2 | 26.0 | 41.0 | 62.0 |
| RootAligned(MCTS) | 98.0 | 98.5 | 98.5 | 79.4 | 81.1 | 81.1 | **99.3** | **99.3** | **99.3** | **83.0** | 85.0 | 85.0 |
| LocalRetro(MCTS) | 92.5 | 94.5 | 95.5 | 44.3 | 50.9 | 58.3 | 86.7 | 90.0 | 95.3 | 52.0 | 55.0 | 62.0 |
| Graph2Edits(Retro*) | 92.0 | 95.5 | 97.5 | 51.1 | 59.4 | 80.0 | 94.0 | 95.0 | 97.5 | 71.0 | 74.0 | 82.0 |
| RootAligned(Retro*)† | **99.0** | 99.0 | 99.0 | **86.8** | 88.9 | 88.9 | 98.7 | 98.7 | 98.7 | 78.0 | 82.0 | 82.0 |
| LocalRetro(Retro*) | 95.5 | 97.5 | 98.0 | 51.0 | 65.8 | 73.7 | 97.3 | 99.3 | 99.3 | 63.0 | 69.0 | 72.0 |
| LLM(MCTS) | 54.5 | 68.5 | 75.5 | 25.8 | 27.2 | 31.3 | 12.7 | 17.3 | 20.7 | 0.0 | 4.0 | 5.0 |
| LLM(Retro*) | 56.0 | 69.0 | 75.5 | 23.2 | 26.8 | 30.6 | 14.7 | 19.3 | 13.3 | 0.0 | 2.0 | 5.0 |
| LLM-Syn-Planner (GPT) | 91.0 | **99.5** | **100.0** | 64.7 | 91.1 | 92.1 | 93.3 | 98.0 | 98.0 | 72.0 | **86.0** | **87.0** |
| LLM-Syn-Planner (DS) | 93.0 | **99.5** | **100.0** | 62.1 | **92.1** | **92.6** | 96.7 | **99.3** | **99.3** | 74.0 | 84.0 | 86.0 |

*Table 2.* Summary of retrosynthesis planning performance across four datasets. The best model for each experiment setting is bolded and the top three are underlined. All runs were limited to 60 minutes per molecule. $N$ denotes the model call limit. We denote LLM-Syn-Planner using GPT-4o as LLM-Syn-Planner (GPT) and LLM-Syn-Planner using DeepSeek-V3 as LLM-Syn-Planner (DS). † The RootAligned model does not finish 300 model calls in 60 minutes due to high computational cost.

of LLM-Syn-Planner in Appendix C.3.

### 4.3. Synthesizable Design

To evaluate the synthesizable design capability of LLM, we first consider common heuristic oracle functions relevant to bioactivity and drug discovery. We compare LLM-Syn-Designer with various molecular optimization methods, including Graph-GA (Jensen, 2019), REINVENT (Olivecrona et al., 2017), MolLEO (Wang et al., 2024), and MARS (Xie et al., 2021), and present the results in Figure 3. Notably, the baseline methods do not enforce synthesizability constraints, allowing them to explore a broader chemical space and achieve higher scores, albeit with non-synthesizable molecules. The results demonstrate that LLM-Syn-Designer effectively balances optimization efficiency and synthesizability. In all cases, the best molecules identified by LLM-Syn-Designer exhibit competitive or superior fitness compared to traditional algorithms and MolLEO, while ensuring synthesizability. Specifically, for the isomers_C9H10N2O2PF2Cl target, LLM-Syn-Designer achieves comparable or higher scaled fitness values with fewer oracle calls than all other methods. This suggests that integrating synthesizability constraints within the optimization process does not necessarily compromise efficiency.

### 4.4. Ablation study

We conducted several ablation studies to evaluate different design choices: route formats, the use of molecule RAG, reward signals, EA parameters, and prompt robustness. The results are shown in Table 3.

**Observation 1: The linear format of synthesis steps significantly outperforms the tree format.** We investigate the influence of route format in Table 3. The results suggest that linear storage of decision steps better reduces the ex-

ponentially growing complexity of the synthesis pathway, thus leading to much higher success rates. Additionally, we introduce a simple baseline named (Textual + Extraction) to allow the LLM to generate in an arbitrary format, followed by a subsequent query to extract the route from the returned response. Surprisingly, this approach also yields decent performance, even with an unconstrained format.

**Observation 2: Even rough intermediate feedback can be significantly useful for LLMs.** To isolate the contribution of the Molecule-RAG module in our retrosynthesis planner, we perform two ablations: (i) removing RAG entirely and (ii) substituting the retrieved routes with random routes in the `INITIALIZATION` and `MUTATION` prompts. Surprisingly, random routes, though not related to the synthesis of the target molecule, still significantly increase performance, indicating that they serve as generic in-context exemplars for the LLM. When we employ RAG with routes retrieved via Morgan-fingerprint similarity, the improvement is even larger. This holds true even though Morgan fingerprints do not directly encode synthetic feasibility and structurally close analogues are not always present in the database. These findings demonstrate that LLMs can extract value from rough, intermediate feedback, and that a lightweight RAG component can markedly enhance retrosynthetic planning quality.

**Observation 3: Partial reward is crucial for long-horizon sequential decision-making.** The target reward is very sparse as it only evaluates if the entire synthesis pathway is valid. We validate the importance of partial reward by introducing a simple synthesis accessibility evaluator (SC score). With partial reward, the success rate improves considerably across both datasets.

**Observation 4: Reinforcing explainability helps improve LLM performances.** We further examined the impact

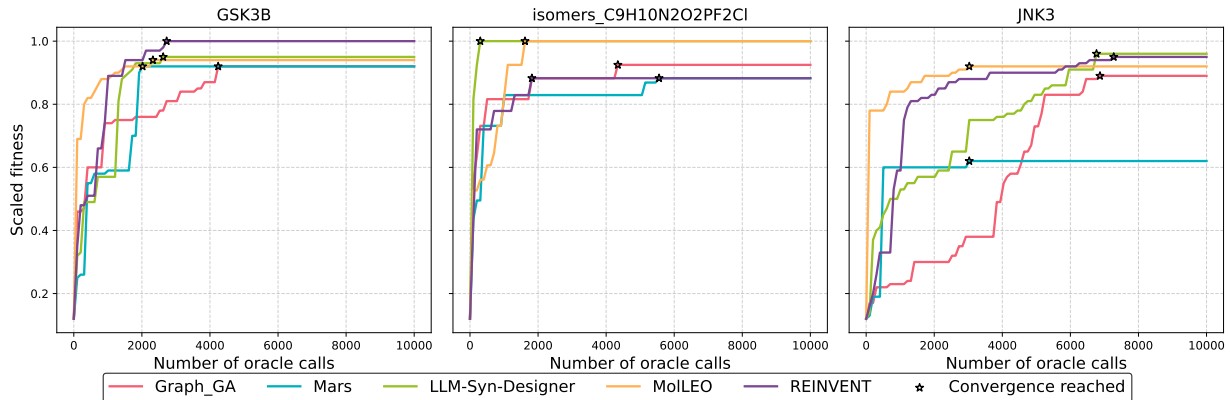

*Figure 3.* Fitness score of the best molecule found by each molecule optimization method. Only LLM-Syn-Designer (GPT) here ensures the synthesizability of the found molecule.

of prompt design. Incorporating an explicit explanation section in the prompt consistently enhanced performance, indicating that exposing the model's intermediate reasoning steps helps the LLM arrive at more accurate decisions.

## 5. Related Work

### 5.1. ML-based Single-step Retrosynthesis Models

Single-step retrosynthesis models predict the outcome of a single reaction step, i.e., given an input molecule, how can it be decomposed and into which constituents? Early works directly predicted precursors by seq-to-seq translation on SMILES (Weininger, 1988; Liu et al., 2017) or using fingerprints (Segler & Waller, 2017; Coley et al., 2017; Fortunato et al., 2020). More recently, single-step retrosynthesis models have employed transformers (Vaswani, 2017) and graph neural networks (GNNs). Methods can be broadly categorized into template-based, template-free, or semi-template methods. Template-based methods use pre-defined chemical rules which can be advantageous if they are defined with high granularity (Szymkuć et al., 2016; Grzybowski et al., 2018; Segler & Waller, 2017; Dai et al., 2019; Ishida et al., 2019; Seidl et al., 2022; Chen & Jung, 2021; Xie et al., 2023). Template-free methods attempt to learn these rules from data and learn a translation (Liu et al., 2017; Zheng et al., 2019; Schwaller et al., 2020; Zhong et al., 2022). Finally, semi-template methods make intermediate predictions (such as synthons) and then predict the precursors based on these (Shi et al., 2020; Somnath et al., 2021; Sacha et al., 2021; Zhong et al., 2023).

### 5.2. Search-directed Retrosynthesis Planning

By coupling a search algorithm with single-step retrosynthesis models, multi-step retrosynthesis can be performed. Exemplary works include applying Monte Carlo tree search (MCTS) (Segler et al., 2018), Retro* (Chen et al., 2020),

| Setting | Variant | USPTO Easy | USPTO-190 |
|---|---|---|---|
| Route Format | Textual + Extraction | 84.5 | 48.9 |
| | Tree | 65.5 | 13.1 |
| | Sequential | 91.0 | 64.7 |
| Retrieval | w/o example routes | 51.0 | 20.0 |
| | w/ random routes | 84.0 | 52.1 |
| | w/ retrieved routes | 91.0 | 64.7 |
| Reward | w/ only final reward | 63.5 | 15.3 |
| | w/ partial reward | 91.0 | 64.7 |
| Prompt | w/o \<Explanation> | 79.5 | 55.3 |
| | w/o 'Rational' | 88.5 | 62.6 |
| | Full | 91.0 | 64.7 |

*Table 3.* Ablation study of LLM-Syn-Planner across different design choices: route format, use of molecule RAG, reward signal, EA parameters and robustness of the prompt. We use GPT-4o as the LLM and all the experiments are conducted under $N = 100$.

Planning with Dual Value Networks (PDVN) (Liu et al., 2023), and a recent double-ended search algorithm (Yu et al., 2024). Since retrosynthesis has broad applicability for molecular discovery, many retrosynthesis platforms exist, encompassing industrial (Szymkuć et al., 2016; Grzybowski et al., 2018; Genheden et al., 2020; Saigiridharan et al., 2024; Watson et al., 2019; Molecule.one; Schwaller et al., 2020) and open-source (Genheden et al., 2020; Saigiridharan et al., 2024; Coley et al., 2019; Tu et al., 2025) solutions. Very recently, works have investigated applying LLMs for retrosynthesis through fine-tuning (Nguyen-Van et al., 2024), instruction-tuning (Yang et al., 2024), platform assistants (Zhang et al., 2025), experimental planning agents (Liu et al., 2024b), and integration with knowledge graphs for synthesis planning of polymers (Ma et al., 2025).

## 6. Conclusion

In this paper, we studied the retrosynthesis problem with LLMs. Specifically, we experimented with using LLMs as

single-step reaction prediction models with a search algorithm and found LLMs significantly underperformed specialized reaction models. To improve this, we proposed to sample entire multi-step synthetic pathways and introduced an evolutionary process to optimize them. To scale this approach, we leveraged a linear format to store reaction steps and designed partial rewards with retrieved reaction sub-trajectories. In the end, we bridged the performance gap and matched the SOTA performance in retrosynthesis planning. In addition, we demonstrated LLMs can be easily adapted to the synthesizable molecular design problem to find property-optimized molecules that are synthesizable.

**Limitation and future work:** Despite promising results, we observed that LLMs suffered significantly with sparse rewards (e.g. in the shooting setup) while improved significantly with partial rewards and retrieved sub-trajectories. It is worth studying how to incorporate a search algorithm into our framework when LLMs struggle to generate any synthesis paths with the desired target molecule. For future work, it is promising to study more flexible design criteria enabled by LLMs such as material-constrained synthesis planning (Guo & Schwaller, 2024b).

## Acknowledgments

We thank Wenhao Gao for helpful discussions on retrosynthesis planning. We thank the anonymous reviewers for their valuable feedback, in particular suggestions to conduct cases studies for recently commercialized molecules.

## Impact Statement

This work investigates how LLMs can be used for retrosynthesis planning and synthesizable molecule design. Both use-cases are applicable to therapeutics and materials design. No molecules were synthesized and experimentally tested so there are no specific societal consequences we feel should be highlighted. However, in the future, the framework, if properly experimentally validated, could have positive societal benefits.

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

# Appendix for LLM-Augmented Chemical Synthesis and Design Decision Programs

## A. Extended Descriptions

### A.1. Dataset Statistics

We show the dataset statistics in Table 4.

| Name | No. of Routes | Avg. Route Length | Avg. SA score | Avg. SC score |
|---|---|---|---|---|
| USPTO Easy | 200 | 3.7 | 2.8 | 3.8 |
| USPTO-190 | 190 | 6.7 | 3.6 | 4.0 |
| Pistachio Reachable | 150 | 5.5 | 3.1 | 3.9 |
| Pistachio Hard | 100 | 7.5 | 3.6 | 3.9 |

*Table 4.* Statistics of the dataset used in the experiments.

### A.2. MCTS for Retrosynthesis Planning

The single-step model predicts potential sets of reactants for a given product, transforming a target molecule into plausible precursors. However, multiple steps may be needed to reach commercially available or easily synthesized materials. This is why the single-step reaction model is integrated with MCTS: it systematically explores these multi-step routes, pruning unlikely paths while focusing on the most promising transformations. By striking a balance between exploration and exploitation, MCTS avoids getting stuck in unproductive branches and can uncover synthetic routes that might not be obvious through manual inspection alone.

Under the MCTS procedure, the target molecule is defined as the root node of a search tree, and each edge represents a single-step retrosynthetic transformation predicted by the reaction model. A policy network can be used to rank or filter the most promising disconnection suggestions at each step, while a value function provides an estimate of how likely a given partial route is to succeed in the long run. The algorithm selects which node to expand next using an Upper Confidence Bound (UCB), which balances the value estimate (exploitation) with the uncertainty in that estimate (exploration). A reward function then quantifies the outcome of each expansion—often based on reaction feasibility, synthetic cost, or reaching known starting materials. These reward signals are backpropagated to update the value estimates of each node. Finally, iterating selection, expansion, simulation, and backpropagation until we reach a termination condition (time limit, enough solutions found).

### A.3. Retro* Algorithm

Retro* (Chen et al., 2020) integrates neural networks with a best-first search strategy to solve retrosynthesis problems. It models the problem as an AND-OR tree, where "AND" nodes represent reactions and "OR" nodes correspond to molecules. A neural network, trained on prior retrosynthesis experiences, estimates the cost of each node. Using a best-first search, the algorithm prioritizes the most promising pathways based on these predictions. It then applies a single-step model to expand the selected node, generating an AND-OR subtree. Finally, it updates the pathway costs to guide the next selection step.

### A.4. Additional Experimental details

For single-step models, we use the checkpoints from syntheseus [3]. In the MCTS algorithm, we employ a basic reward function: a state receives a reward of 1.0 if all molecules are purchasable (i.e., the state is solved), and 0.0 otherwise. The

---

[3]https://github.com/microsoft/syntheseus

value function is set as a constant 0.5. For policy, we use softmax values derived from the single-step reaction model, scaled by a temperature of 3.0 and normalized across the total number of reactions.

In the Retro* algorithm, we follow the retro*-0 variant described in the original paper (Chen et al., 2020). The OrNode cost function assigns a cost of 0 to purchasable molecules and infinity otherwise. The AndNode cost function defines the reaction cost as -log(softmax) of the reaction model output, thresholded at a minimum value. For the search heuristic (value function), we use a constant value of 0, consistent with the retro*-0 algorithm.

### A.5. Cost of using LLMs

Indeed, large LLMs such as GPT-4 currently require more computing than traditional models like RootedAligned. However, our motivation is to rigorously examine what LLMs uniquely offer in complex, high-level scientific reasoning tasks like multi-step retrosynthesis planning—a setting where domain-specific models often require extensive retraining and are limited in adaptability.

Our results demonstrate that even without any fine-tuning, LLM-Syn-Planner matches or outperforms specialized models across multiple datasets. This zero-shot capability highlights a crucial point: LLMs offer general-purpose reasoning and chemical adaptability out-of-the-box, which cannot be achieved by most lightweight models without costly re-engineering when reaction databases or design goals change.

Furthermore, while cost is a valid concern, we believe it must be evaluated in context:

▷ The retraining and maintenance overhead for specialized models is non-trivial in dynamic research environments.

▷ Our LLM-based system can immediately leverage new knowledge via retrieval, without retraining.

▷ LLM inference costs are rapidly decreasing as optimized deployment (e.g., quantization, distillation, smaller expert LLMs) becomes mainstream.

▷ As open-source LLMs improve, smaller models will become more capable (this can also be achieved via distillation). These small models can be hosted locally, thus also not requiring large clusters to host (for example, using Ollama to host DeepSeek R1).

Lastly, LLM-Syn-Planner represents a first step toward a broader vision of flexible, generalist AI for scientific discovery—something static models cannot enable. While not yet universally cost-effective, we argue that the emerging capabilities and flexibility of LLMs justify this early-stage investigation.

### A.6. Computational Resources

Our experiments utilized the GPT-4o model and the DeepSeek-V3 model. The GPT-4o model refers to the `GPT-4o` checkpoint from 2024-08-06 [4]. All GPT-4o checkpoints were hosted on Microsoft Azure[5].

## B. Extended Related Work

### B.1. Synthesizable Molecular Design

Synthesizable molecular design aims to generate molecules (with optimal properties) that are also synthesizable, as predicted by a retrosynthesis model. While retrosynthesis methods are often described as "top-down" because they decompose a target molecule into purchasable precursors, the most common methods in literature for synthesizable molecular design proceeds "bottom-up", which combine building blocks to construct the final molecule. Therefore, instead of predicting the resulting *precursors* from an input molecule, "bottom-up" approaches require a way to predict the *product molecule* given precursors. To this end, existing approaches either use forward synthesis prediction models (Bradshaw et al., 2019; 2020) or define a set of templates which dictate *how* building blocks can be combined (Gao et al., 2022; 2024; Luo et al., 2024; Koziarski et al., 2024; Cretu et al., 2024; Seo et al., 2024; Swanson et al., 2024; Jocys et al., 2024). These methods can be broadly classified as synthesizability-constrained generative models. An alternative approach is to couple retrosynthesis models directly into

---

[4].https://platform.openai.com/docs/models

[5] *.openai.azure.com

| Method | LLM | USPTO Easy | Pistachio Hard |
|---|---|---|---|
| Direct query | GPT-4o | 4.0 | 0.0 |
| | DeepSeek-V3 | 4.5 | 1.0 |
| LLM-Syn-Planner | GPT-4o | 91.0 | 72.0 |
| | DeepSeek-V3 | 93.0 | 74.0 |

*Table 5.* Ablation studies of retrosynthesis planning based on direct user queries. We report the Solve rate under $N = 100$.

the optimization loop of generative models, such that synthesizability is optimized for, rather than enforced in the generation process (Guo & Schwaller, 2024a;b).

## C. Extended Experiment Results

### C.1. Performance of direct user queries for multi-step retrosynthesis tasks

In Table 5, we show the performance of directly querying LLMs with target molecules. Each LLM was queried 100 times to ensure a fair comparison. On the USPTO easy dataset, both LLMs solved fewer than 10 routes, and on the Pistachio Hard dataset, they failed in nearly all cases. These results suggest the models do not merely "remember" synthetic routes from their training data.

### C.2. Performance of advanced reasoning LLM (DeepSeek-R1)

| LLM | USPTO Easy | USPTO-190 |
|---|---|---|
| GPT-4o | 91.0 | 64.7 |
| DeepSeek-V3 | 93.0 | 62.1 |
| DeepSeek-R1 | 95.0 | 57.9 |

*Table 6.* Ablation studies of LLM-Syn-Planner using DeepSeek-R1. We report the Solve rate under $N = 100$.

We conduct ablation studies on advanced reasoning LLM DeepSeek-R1 (Guo et al., 2025) and show the results in Table 6. Interestingly, although DeepSeek-R1 is designed as a reasoning model, it performs worse than DeepSeek V3. Moreover, because DeepSeek-R1 includes its thinking process in the output, its overall cost is roughly three times higher than that of DeepSeek-V3.

### C.3. Case Study

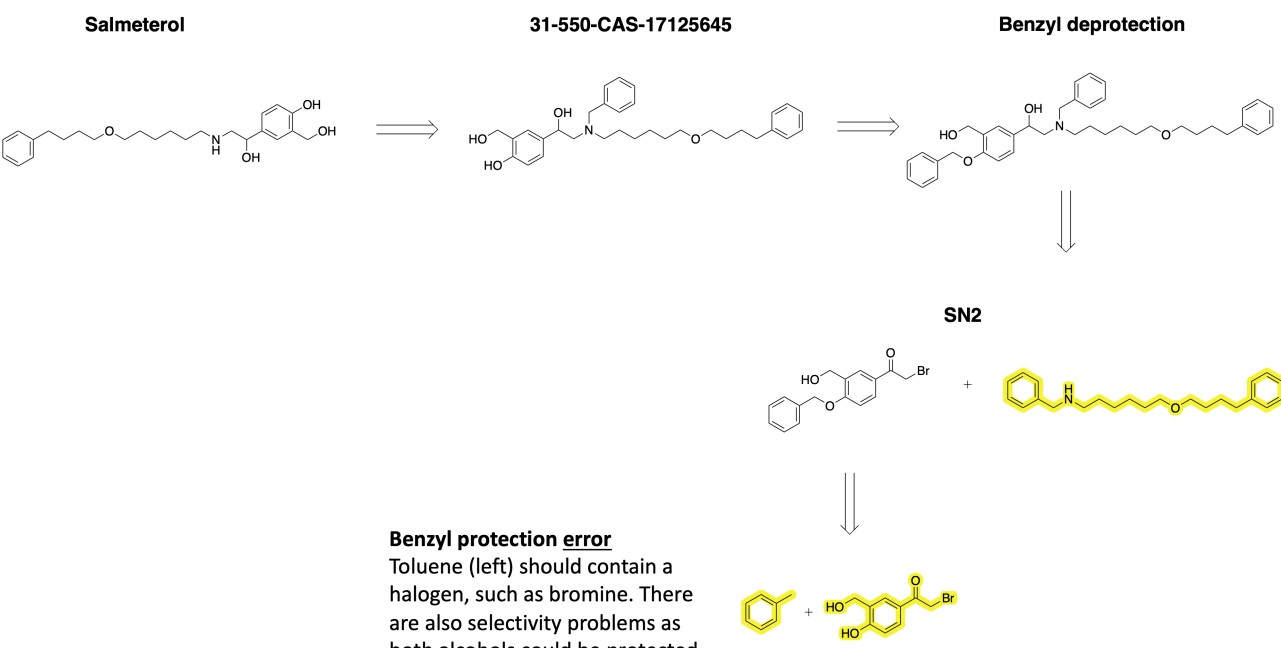

*Figure 4.* Proposed synthesis route to Salmeterol. Available building blocks are highlighted in yellow.

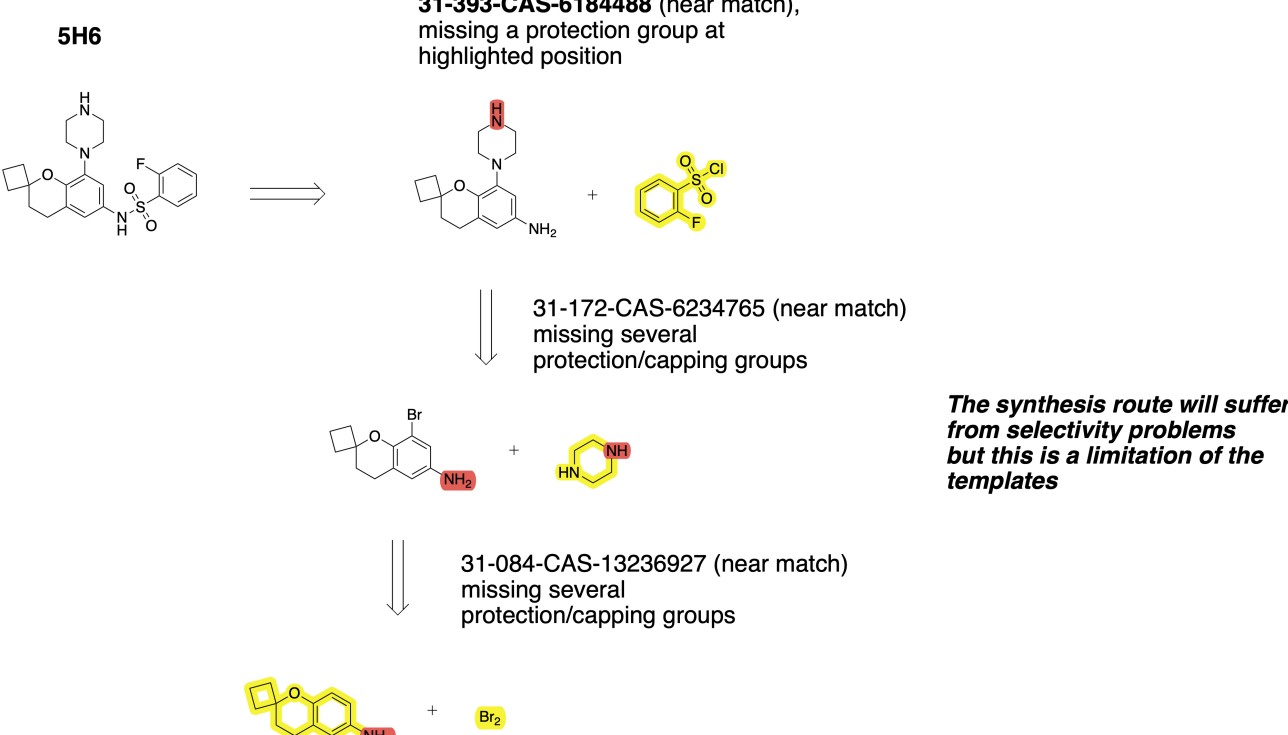

*Figure 5.* Proposed synthesis route to 5H6. Available building blocks are highlighted in yellow.

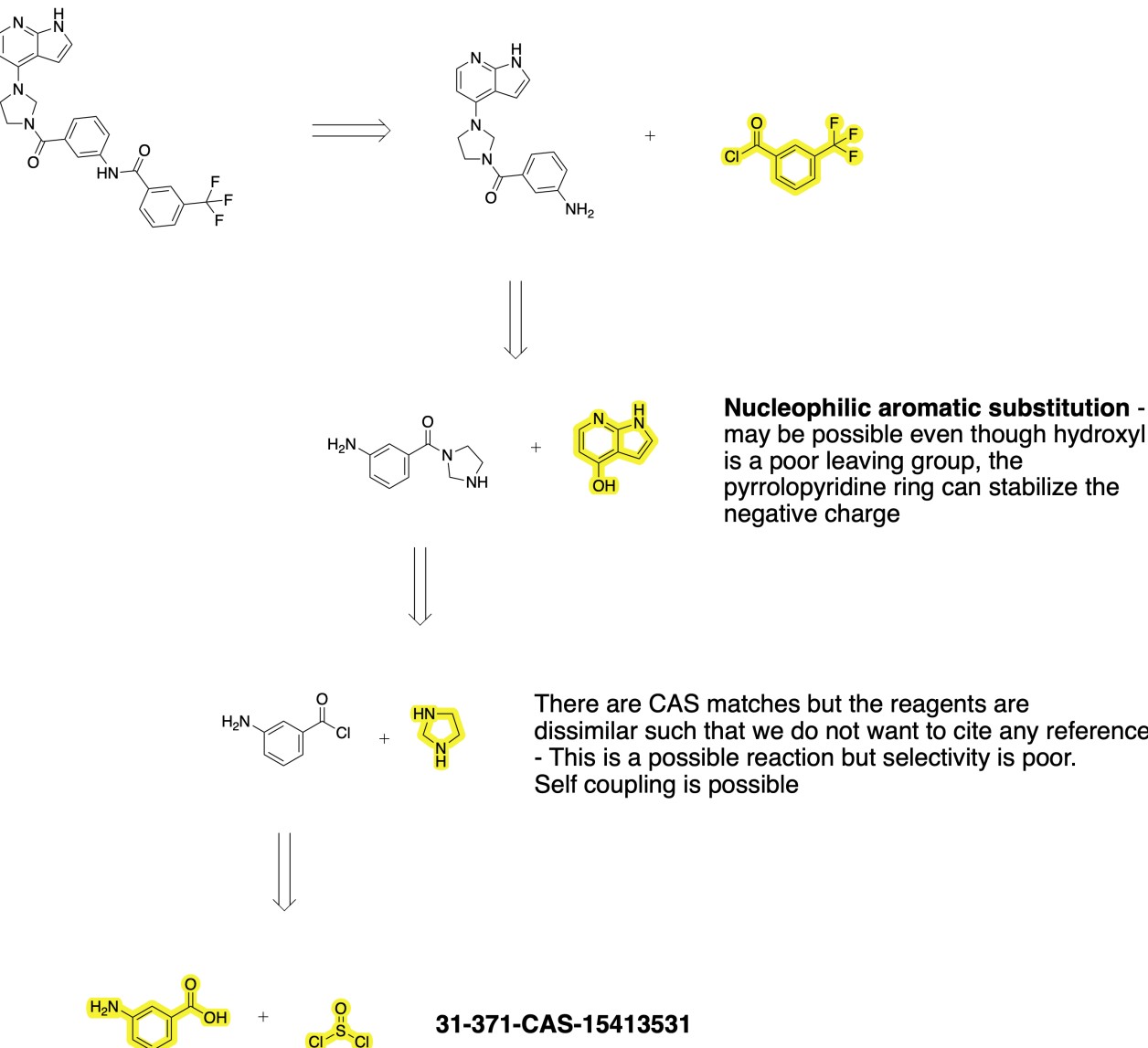

*Figure 6.* Proposed synthesis route to a DDR1 inhibitor. Available building blocks are highlighted in yellow.

*Figure 7.* Proposed synthesis route to Lenalidomode. Available building blocks are highlighted in yellow.

We apply LLM-Syn-Planner to propose synthetic routes to four bio-active molecules: **Salmeterol** (Figure 4), **5H6** (Figure 5), **DDR1 inhibitor** (Figure 6), and **Lenalidomide** (Figure 7). For all molecules, a synthetic route combining the fixed templates and building blocks stock is successfully proposed. To assess the feasibility of the proposed reaction sequences, we look for literature precedent using SciFinder (Gabrielson, 2018; CAS) and annotate the CAS number for matched reaction steps. For reaction steps without a literature reference, we propose a plausible reaction transformation. Lastly, and most importantly, we highlight reactivity and selectivity problems across all routes for transparency. Since LLM-Syn-Planner uses a fixed template set, proposed synthetic routes inherit the limitations of the templates. Consequently, for all synthetic routes, there is at least one instance of a reactivity problem which would likely involve modifying the route if it were to be performed in the lab. With improved templates, the chemical reliability of LLM-Syn-Planner will improve.

## D. Prompts

We show the prompts of INITIALIZATION and MUTATION for LLM-Syn-Planner. And LLM operators prompt for LLM-Syn-Designer.

---

LLM-Syn-Planner INITIALIZATION prompts

```
As a professional chemist specialized in synthesis analysis, you are tasked with generating a retrosynthesis
route for a target molecule provided in SMILES format.

A retrosynthesis route is a series of retrosynthesis steps that starts from the target molecule and ends
with some commercially purchasable compounds.  The reactions are from the USPTO dataset.  Please also
```

---

**Top 1 from LLM-syn-designer, JNK3 = 0.96**

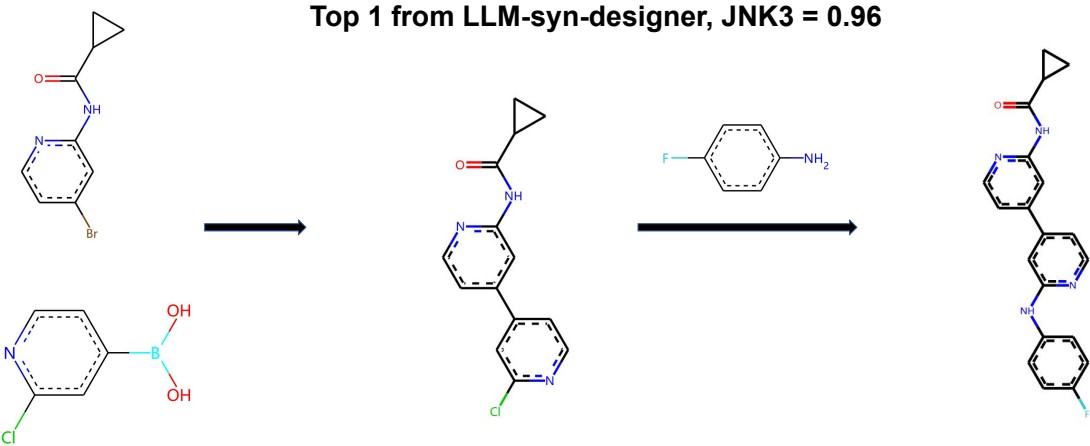

*Figure 8.* **Top 1 molecule of jnk3 found by LLM-Syn-Designer.**

```
consider reactions in stereochemistry.

The route should be a list of steps wrapped in <ROUTE></ROUTE> with < EXPLANATION></EXPLANATION> after
it.  Each step in the list should be a dictionary.  You need to keep a molecule set, which consists of the
molecules we need to synthesize or purchase.  In each step, you need to select a molecule from the 'Molecule
set' as the product molecule in this step and use a backward reaction to find the reactants.  After taking
the backward reaction in this step, you need to remove the product molecule from the molecule set and add
the reactants you find into the molecule set, and then name this updated set as the 'Updated molecule set'
in this step.  In the next step, the starting molecule set should be the 'Updated molecule set' from the
previous step.  In the last step, all the molecules in the 'Updated molecule set' should be purchasable.
Here is an example:

<ROUTE>
    [
        {
            'Molecule set': "[Target Molecule]",
            'Rational': Step analysis,
            'Product': "[Product molecule]",
            'Reaction': "[Reaction template]",
            'Reactants': "[Reactant1, Reactant2]",
            'Updated molecule set': "[Reactant1, Reactant2]"
        },
        {
            'Molecule set': "[Reactant1, Reactant2]",
            'Rational': Step analysis,
            'Product': "[Product molecule]",
            'Reaction': "[Reaction template]",
            'Reactants': "[subReactant1, subReactant2]",
            'Updated molecule set': "[Reactant1, subReactant1, subReactant2]"
        }
    ]
</ROUTE>
<EXPLANATION>: Explanation for the whole route. </EXPLANATION>\\

Requirements:
1.  The 'Molecule set' contains molecules we need to synthesize at this stage.  In the first step, it should
be the target molecule.  In the following steps, it should be the 'Updated molecule set' from the previous
step.
2.  The 'Rational' part in each step should be your analysis for synthesis planning in this step.  It should
be in the string format wrapped with '  '.
3.  'Product' is the molecule we plan to synthesize in this step.  It should be from the 'Molecule set'.
The molecule should be a molecule from the 'Molecule set' in a list.  The molecule smiles should be wrapped
with '  '.
4.  'Reaction' is a reaction that can synthesize the product molecule.  It should be on a list.  The
reaction template should be in SMILES format.  For example, [Product»Reactant1.Reactant2].
5.  'Reactants' are the reactants of the reaction.  It should be on a list.  The molecule smiles should be
wrapped with '  '.
6.  The 'Updated molecule set' should be molecules we need to purchase or synthesize after taking this
reaction.  To get the 'Updated molecule set', you need to remove the product molecule from the 'Molecule
```

set' and then add the reactants in this step into it.  In the last step, all the molecules in the 'Updated molecule set' should be purchasable.
7.  In the <EXPLANATION>, you should analyze the whole route and ensure the molecules in the 'Updated molecule set' in the last step are all purchasable.
My target molecule is:
{Target Molecule}
To assist you, example retrosynthesis routes that are either close to the target molecule or representative will be provided.

<ROUTE>
Retrieved route here
</ROUTE>

Please propose a retrosynthesis route for my target molecule.  The provided reference routes may be helpful.
You can also design a synthetic route based on your own knowledge.

## LLM-Syn-Planner MUTATION prompts

As a professional chemist specializing in synthesis analysis, you are tasked with modifying a retrosynthesis route for target molecules provided in SMILES format.
A retrosynthesis route is a series of retrosynthesis steps that starts from the given target molecule set and ends with some commercially purchasable compounds.  In the route, you need to keep a molecule set, which are the molecules we need.  In the first step, the molecule set should be the target molecule set given by the user.  In each step, you need to provide a backward reaction and update the molecule set.  Specifically, you need to remove the product molecule of the reaction from the molecule set and then add the reactants to it.
By doing so, you will end with a molecule set in which all the molecules are commercially purchasable.  The reactions are from the USPTO dataset.  Please also take reactions in stereochemistry into consideration.
For example, E-configuration or Z-configuration.
The route should be a list of steps wrapped in <ROUTE></ROUTE> with <EXPLAINATION></EXPLAINATION> after it.  Each step in the list should be a dictionary.  You need to keep a molecule set, which consists of the molecules we need to synthesize or purchase.  In each step, you need to select a molecule from the 'Molecule set' as the product molecule in this step and use a backward reaction to find the reactants.  After taking the backward reaction in this step, you need to remove the product molecule from the molecule set add the reactants you find into the molecule set, and then name this updated set as the 'Updated molecule set' in this step.  In the next step, the starting molecule set should be the 'Updated molecule set' from the previous step.  In the last step, all the molecules in the 'Updated molecule set' should be purchasable.
Here is an example:

```
<ROUTE>
    [
        {
            'Molecule set': "[Target Molecule]",
            'Rational': Step analysis,
            'Product': "[Product molecule]",
            'Reaction': "[Reaction template]",
            'Reactants': "[Reactant1, Reactant2]",
            'Updated molecule set': "[Reactant1, Reactant2]"
        },
        {
            'Molecule set': "[Reactant1, Reactant2]",
            'Rational': Step analysis,
            'Product': "[Product molecule]",
            'Reaction': "[Reaction template]",
            'Reactants': "[subReactant1, subReactant2]",
            'Updated molecule set': "[Reactant1, subReactant1, subReactant2]"
        }
    ]
</ROUTE>
<EXPLANATION>: Explanation for the whole route. </EXPLANATION>\\
```

Requirements:
1.  The 'Molecule set' contains molecules we need to synthesize at this stage.  In the first step, it should be the target molecule set.  In the following steps, it should be the 'Updated molecule set' from the previous step.
2.  The 'Rational' part in each step should be your analysis for synthesis planning in this step.  It should be in the string format wrapped with ' '
3.  'Product' is the molecule we plan to synthesize in this step.  It should be from the 'Molecule set'.
The molecule should be a molecule from the 'Molecule set' in a list.  The molecule smiles should be wrapped with ' '.
4.  'Reaction' is a reaction that can synthesize the product molecule.  It should be on a list.  The reaction template should be in SMILES format.  For example, [Product»Reactant1.Reactant2].
5.  'Reactants' are the reactants of the reaction.  It should be on a list.  The molecule smiles should be wrapped with ' '.
6.  The 'Updated molecule set' should be molecules we need to purchase or synthesize after taking this reaction.  To get the 'Updated molecule set', you need to remove the product molecule from the 'Molecule set' and then add the reactants in this step into it.  In the last step, all the molecules in the 'Updated

```
molecule set' should be purchasable.
7.  In the <EXPLANATION>, you should analyze the whole route and ensure the molecules in the 'Updated
molecule set' in the last step are all purchasable.

My target molecule set is:
{Target Molecule set}
Here is the feedback for the route:
{Feedback}
To assist you, example retrosynthesis routes that are close to the target molecules in the starting molecule
set will be provided.

<ROUTE>
Retrieved route here
</ROUTE>

Please propose a retrosynthesis route for the starting molecule set.  The provided reference routes may be
helpful.  You can also design a synthetic route based on your own knowledge.  All the molecules should be
in SMILES format.  For example, Cl2 should be ClCl in SMILES format.  Br2 should be BrBr in SMILES format.
H2O should be O in SMILES format.  HBr should be [H]Br in SMILES format.  NH3 should be N in SMILES format.
Hydrogen atoms are implicitly understood unless explicit clarity is needed.
```

## LLM-Syn-Designer prompts

```
I have two molecules and their JNK3 scores.  The JNK3 score measures a molecularś biological activity
against JNK3.
Molecule 1 SMILES, Molecule 1 score
Molecule 2 SMILES, Molecule 2 score

Now I want to synthesize a new molecule that has a higher JNK3 score.  Please propose a new synthesizable
molecule that has a higher JNK3 score.  You can either make crossovers and mutations based on the given
molecules or just propose a new molecule based on your knowledge.

Your output should follow the format:

<EXPLANATION>Your analysis</EXPLANATION>
 <MOLECULE>The SMILES of your proposed molecule</MOLECULE>

Here are the requirements:
1.  In the <EXPLANATION>, you should analyze how to edit the given molecules to get a better property
score and then propose your edited molecule or your proposed new molecule, and how to synthesize your
proposed/edited molecule.
2.  In the <MOLECULE>, you should provide the SMILES of the molecule you propose.
```

