# OpenReview forum: "LLM-Augmented Chemical Synthesis and Design Decision Programs"
_ICML.cc/2025/Conference — ICML 2025 poster_

### Official Review · Reviewer_cmv4 · 2025-02-21

**Overall Recommendation:** 3

**Summary:**

This paper attempts to address the problem of multi-step retrosynthesis planning using large language models (LLMs). The authors first propose a format for representing a synthesis pathway, using a sequential reaction step format to characterize the synthesis pathway. This approach allows large models to avoid the traditional difficulties associated with understanding deeply nested tree structures. Additionally, the authors introduce a generation + refinement pipeline, which involves first generating a complete synthesis pathway in one go and then using three different levels of rewards + prompts to guide the LLM to further evolve the generated pathway, ultimately obtaining a correct synthesis pathway. The authors conducted experimental validations on four subsets of synthesis pathways derived from the USPTO and Pistachio databases. They also applied the proposed method to molecular property optimization, aiming to ensure the synthesizability of molecules while optimizing their properties. Experiments were carried out for three different properties.

**Claims And Evidence:**

Yes

**Essential References Not Discussed:**

In this submission, the design of synthesizable molecules is discussed, and the authors' model is compared with other molecule optimization methods, claiming that their model can optimize properties while ensuring synthesizability, which other baselines cannot achieve. However, an excellent work [1] appeared at ICML 2024 that converts unsynthesizable molecules into synthesizable ones while as much as possible ensuring the molecular properties. By using this ICML-published article to optimize the synthesizability of all baseline-optimized molecules, both the properties and synthesizability of the molecules would be ensured to some extent. The submission did not design any experiments to demonstrate that the proposed method could achieve better results than calling the molecule optimization model and this ICML 2024 work in series.

**Reference**

[1] Luo S, Gao W, Wu Z, et al. Projecting Molecules into Synthesizable Chemical Spaces[C]//International Conference on Machine Learning. PMLR, 2024: 33289-33304.


#

**Experimental Designs Or Analyses:**

I have checked all the experimental designs and analyses. Here are the problems:

- for retrosynthesis planning

  - When using RootAligned for single-step retrosynthesis prediction, did you employ test-time augmentation? Although this can improve the solve rate, it significantly reduces the model's response speed. The original paper of RootAligned[3] set the number of test-time augmentation to 20, meaning that the same sample would undergo 20 inferences. If you reduce this parameter to 10 or 5, you might be able to increase the number of model responses per unit time at the cost of a slight decrease in the accuracy of single-step retrosynthesis prediction.

  - Setting a maximum solving time is not fair, as GPT-4o is deployed on a very powerful cluster while the baselines are not. The authors should deploy these baseline models on a server as powerful as the OpenAI cluster and then set a unified maximum solving time; otherwise, it is unfair.

  - The datasets used in the experiments are relatively old, and most of them are accompanied by detailed original papers and patent articles. These articles may have already become part of the pretraining corpus for GPT-4o. It is difficult for the authors to prove whether the synthesis path proposed by the model has been seen during pretraining. The authors should present some recently marketed or complex molecules as case studies to demonstrate that the proposed method truly works. The following are some convincing choices:

    - Pacritinib, SMILES: `C1=CC2=CC(=C1)COCC=CCOCC1=C(OCCN3CCCC3)C=CC(=C1)NC1=NC=CC2=N1`
    - Daprodustat, SMILES `O=C(O)CNC(=O)C1C(=O)N(C2CCCCC2)C(=O)N(C2CCCCC2)C1=O`
    - Nirmatrelvir, SMILES `C(N[C@@H](C[C@H]1C(=O)NCC1)C#N)(=O)[C@@H]2[C@@]3([C@@](C3(C)C)(CN2C([C@@H](NC(C(F)(F)F)=O)[C@](C)(C)C)=O)[H])[H]`
    - 5-HT6 receptor, SMILES `O=S(=O)(NC=1C=C(C=2OC3(CCC2C1)CCC3)N4CCNCC4)C=5C=CC=CC5F`

    These molecules have been used as case studies in references [1, 2, 3], and all of them have been predicted with reasonable synthesis routes by single-step retrosynthesis models combined with manual expansion by experts. The reviewer believes that these cases are molecules that AI can accurately predict synthesis routes for but are still challenging.

- for synthesizable molecular design

  - The proposed method only compared some molecular optimization methods, but these methods do not guarantee the synthesizability of the optimized molecules. In fact, there have already been works on improving the synthesizability of molecules without compromising their properties (see **Essential References Not Discussed**). Experiments and analyses about synthesizable molecular design need to combine molecular property optimization with molecular synthesizability optimization as a baseline, instead of unilaterally claiming that molecular property optimization methods cannot guarantee synthesizability and thus the proposed method is better.

**Reference**

[1] Yao L, Guo W, Wang Z, et al. Node-aligned graph-to-graph: Elevating template-free deep learning approaches in single-step retrosynthesis[J]. JACS Au, 2024, 4(3): 992-1003.

[2] Zeng K , Yang B , Zhao X ,et al.Ualign: pushing the limit of template-free retrosynthesis prediction with unsupervised SMILES alignment[J].Journal of Cheminformatics, 2024, 16(1):1-16.DOI:10.1186/s13321-024-00877-2.

[3] Zhong Z, Song J, Feng Z, et al. Root-aligned SMILES: a tight representation for chemical reaction prediction[J]. Chemical Science, 2022, 13(31): 9023-9034.

**Methods And Evaluation Criteria:**

Yes

**Other Comments Or Suggestions:**

I have no other comments.

**Other Strengths And Weaknesses:**

**Strengths**

This article pioneers an innovative approach, namely the idea of generating a complete synthesis pathway in one go and then refining it. This could potentially introduce a new paradigm for future multi-step retrosynthesis planning.

**Weakness**

In the sequential structure formatting of the synthesis pathway designed in the article, the size of the molecule set in each step will increase linearly with the number of steps. Consequently, the total number of tokens in the entire synthesis pathway will have an almost quadratic relationship with the number of steps. This is clearly not concise.

**Questions For Authors:**

1. Why is it necessary to replace the chemical reaction proposed by the LLM with a retrieved one during the **Evaluation** process? Does this operation alter the updated molecule set of the replaced step and the subsequent synthesis steps? **It is particularly important to note that  the reaction database is constructed from USPTO-FULL, which contains all the single-step reactions of the routes in USPTO Easy and USPTO-190. If such a replacement is actually carried out, the method would involve the problem of label leakage.**
2. How does the paper verify the **feasibility** of the synthesis pathway when calculating the solve rate? Does it check whether the reactants of each step can produce the desired product through the reaction?
3. What are the **practical advantages** of the proposed method? Retrosynthesis planning is a very application-driven topic. The proposed method does not outperform specialized models in terms of performance, and it relies on large-scale computing power (needing to call powerful LLM APIs) and a pre-defined reaction database, while specialized models can run under more modest computing conditions (for example, models like localRetro can perform a single-step retrosynthesis prediction within a second on a 1080ti graphics card) without the need for a pre-defined reaction database.

If the authors can address all the issues proposed above and the issues in **Experimental Designs Or Analyses**, I will be willing to raise my score.

**Relation To Broader Scientific Literature:**

The idea proposed in this paper of using prompts to generate a synthesis pathway in one go and then optimize it is groundbreaking in multi-step retrosynthesis, and is distinct from other previous works on multi-step retrosynthesis planning.

**Theoretical Claims:**

This paper does not involve any theoretical claims

---

> ### Author Rebuttal · Authors · 2025-04-01
>
> We are extremely appreciative of the reviewer’s feedback and suggestions to improve our work. We are glad that the reviewer considers our work to “potentially introduce a new paradigm for future multi-step retrosynthesis planning”.
>
> We provide additional experiment results in the following link: https://drive.google.com/file/d/1RIcdPo-uV5Fibbllk2COJl9niUhrVAFk/view?usp=sharing
>
> >fair comparison
>
> We follow the Rootaligned paper to set the number of test-time augmentation to 20. In updated Table 1, we extend the search time for each molecule to 60 minutes. Only Rootaligned cannot finish all calls. Due to the time constraint for rebuttal, we will adjust the test-time augmentation in our future version. Note we do not intend to limit the solving time for comparison benefit, but only to rule out methods with extremely long solving times.
>
> >whether the synthesis path proposed by the model has been seen during pretraining. Experiments to show the routes for the provided molecules.
>
> We conducted an ablation study for direct user queries and presented the results in Table 3. The results suggest that the models do not simply “remember” synthetic routes from their training data. We provide the routes for four suggested molecules discovered by our method in the PDF.
>
> >for synthesizable molecular design
>
> In contrast to approaches like [1] that start with an unsynthesizable molecule and then “project” it to a synthesizable form—often compromising similarity and potentially degrading properties (For instance, Table 2 in [1] observes that the average similarity can be modest)—our method jointly optimizes molecular properties and ensures synthesizability from the outset. By directly sampling candidate molecules with high properties and verifying their synthetic accessibility through route sampling, we avoid an additional “projection” step and minimize property loss. We will discuss [1] in our related work as an alternative way to find optimized molecules with synthesis pathways.
>
> > The size of the molecule set will increase
>
> While it is true that the set can increase with each additional step, our framework mitigates this issue in practice. Specifically, during mutation, we provide the LLM only with the partial route starting from the first invalid step, rather than resupplying the entire route. Furthermore, purchasable molecules in the set can be filtered out and do not need to be reintroduced to the LLM. Our experiments indicate that this approach remains manageable in real-world scenarios.
>
> >Replace the chemical reaction proposed by the LLM with a retrieved one. Label leakage.
>
> This procedure does not introduce label leakage because we rely only on reactions known to exist (i.e., those in the database). In fact, the single-step specialized models are also trained on reactions extracted from USPTO-FULL. In the original Retro* paper, they even filter the test route dataset by only keeping the routes whose reactions are all covered by the top 50 predictions by the single-step model (Section 5.1.1).
>
> The main role of retrieving is to use as a template and approximate information as a reference.
>
> > How to verify the feasibility of the synthesis pathway
>
> Yes. Since the reaction templates used are from reliable databases, we can apply the reaction template using the chemical tools rdchiral to verify.
>
> > Practical advantages of the proposed method
>
> Through detailed analysis of the failure cases, we observed that the main issue is not from the LLM itself but from the partial reward mechanism—namely, that optimal SCScore does not guarantee a molecule is in the purchasable set. Consequently, the algorithm can get stuck in an infinite loop.
>
> To address this, we simply added a penalty for repeated entries, which substantially improved performance and, as shown in Table 1, allowed LLM-Syn-Planner to outperform baselines. This indicates that LLMs offer a powerful framework for complex, multi-step decision-making in scientific domains, and we are the first to show that LLMs can achieve comparable or even better performance compared with specialized models. This also provides evidence for solving other science problems with a similar structure.
>
> Regarding the reaction database, it is worth noting that specialized models also require such databases: for instance, localRetro formulates one-step retrosynthesis as a classification problem, necessitating a pre-defined set of reaction data.
>
> At last, we would be happy to engage in further discussions with the reviewer regarding these topics or any additional concerns the reviewer has – please let us know! We thank the reviewer again for their time spent with our paper. If the reviewer finds that our new experiments, analyses, and discussions are useful for improving the paper, we would be grateful if the reviewer would consider a fresher evaluation of our paper.
>
> [1] Luo S, et al. Projecting Molecules into Synthesizable Chemical Spaces, ICML2024
>
> [2] Chen B, et al. Retro*..., ICML2020

---

> > ### Comment · Reviewer_cmv4 · 2025-04-02
> >
> > I appreciate the efforts made by the authors during the rebuttal phase, but the rebuttal does not fully address my concerns. Here are the key points:
> >
> > 1. The authors still fail to adequately address the issue of fairness, or more precisely, the cost-effectiveness problem. Large models like GPT-4, with their massive parameters, require deployment on large-scale clusters to function, while methods like RootedAlign SMILES can operate even on a standard laptop. The authors have not convincingly demonstrated why we need to incur such a high deployment cost to use LLMs for retrosynthesis planning.
> >
> > 2. The rebuttal still contains a fundamental misunderstanding: rdchiral is not a tool for verifying the correctness of reactions. It is specifically designed to correct stereochemistry annotation errors in template-based retrosynthesis predictions.
> >
> > 3. I verified the four proposed pathways using the widely recognized chemistry reaction and literature database available at [SciFinder®](https://scifinder-n.cas.org/). None of the proposed routes could be found in the dataset. Moreover, even the first step of each synthetic route lacks supporting literature. This result demonstrates that the proposed method is not well-suited for real-world applications, especially when the routes of the provided samples can be identified using single-step prediction models paired with simple search methods.
> >
> > Given these reasons, I believe that as an application-driven machine learning submission, this paper fails to demonstrate its advantages in practical applications. Therefore, I choose to maintain my score as **weak reject**, with a negative stance.
> >
> >
> > **Update**
> > 1. I acknowledge the authors' assertion that the advantage of LLMs lies in their ability to avoid retraining, which is particularly beneficial for smaller companies that may lack sufficient data resources.
> >
> > 2. While rdchiral can verify whether reactants can evolve into products based on predefined reaction templates, this approach does not effectively validate the chemical feasibility of the reaction in practice, i.e., whether the reaction templates are chemically valid. This remains a highly challenging issue. However, I agree that the verification method aligns with prior work, which is acceptable.
> >
> > 3. I reserve judgment on whether the four cases I provided are truly "hard cases." However, I would like to emphasize that the baselines listed rely solely on templates from the USPTO-50K dataset (although RetroKNN experiments were conducted on USPTO-Full, I am uncertain whether the authors utilized the corresponding checkpoint). With such a limited template library, it is difficult to argue that the model demonstrates robust reasoning capabilities. As this work is template-based, it shares the same limitations as other template-based approaches. I believe it would be beneficial to investigate the model's performance when exposed to a more comprehensive template library.
> >
> > Based on the above rebuttal, I am willing to revise my score upward to 3.

---

> > > ### Author Response · Authors · 2025-04-04
> > >
> > > Dear reviewer,
> > >
> > > Thanks for your comment. We are happy to discuss your remaining concerns here
> > >
> > > >High deployment cost to use LLMs
> > >
> > > We appreciate the reviewer’s concern regarding deployment cost and fairness. Indeed, large LLMs such as GPT-4 currently require more computing than traditional models like RootedAligned. However, our motivation is to rigorously examine what LLMs uniquely offer in complex, high-level scientific reasoning tasks like multi-step retrosynthesis planning—a setting where domain-specific models often require extensive retraining and are limited in adaptability.
> > >
> > > Our results demonstrate that even without any fine-tuning, LLM-Syn-Planner matches or outperforms specialized models across multiple datasets. This zero-shot capability highlights a crucial point: LLMs offer general-purpose reasoning and chemical adaptability out-of-the-box, which cannot be achieved by most lightweight models without costly re-engineering when reaction databases or design goals change.
> > >
> > > Furthermore, while cost is a valid concern, we believe it must be evaluated in context:
> > >
> > > 1. The retraining and maintenance overhead for specialized models is non-trivial in dynamic research environments.
> > >
> > > 2. Our LLM-based system can immediately leverage new knowledge via retrieval, without retraining.
> > >
> > > 3. LLM inference costs are rapidly decreasing as optimized deployment (e.g., quantization, distillation, smaller expert LLMs) becomes mainstream.
> > >
> > > 4. As open-source LLMs improve, smaller models will become more capable (this can also be achieved via distillation). These small models can be hosted locally, thus also not requiring large clusters to host (for example using Ollama to host DeepSeek R1).
> > >
> > > Lastly, as highlighted by Reviewer kcQx, LLM-Syn-Planner represents a first step toward a broader vision of flexible, generalist AI for scientific discovery—something static models cannot enable. While not yet universally cost-effective, we argue that the emerging capabilities and flexibility of LLMs justify this early-stage investigation.
> > >
> > > >Misunderstanding of rdchiral
> > >
> > > We clarify that the reviewer’s original question was whether we check if reactants can produce the desired product at each step when computing the solve rate. The answer is yes—we use the rdchiralRun function to apply the reaction template and verify if the reactants of each step can produce the desired product through the reaction.
> > >
> > > This is consistent with prior template-based methods [1, 2, 3]. We did not additionally validate the correctness of the reaction templates themselves. As is standard in retrosynthesis planning, we assume that reactions in the database are correct, given that the training of single-step models in all baselines relies on this assumption.
> > >
> > > >Real-world applications
> > >
> > > Thank you for your thorough verification. We would like to clarify that the four molecules selected are challenging cases from prior template-free retrosynthesis work. For Pacritinib, Daprodustat, and Nirmatrelvir, even strong template-based methods (LocalRetro, RetroKNN, MHNreact with MCTS/Retro*) failed to find routes with literature support—indicating the limited coverage of current reaction template sets in USPTO.
> > >
> > > For the 5-HT6 receptor, one of the LLM-Syn-Planner proposed routes is supported by the literature. Additionally, on three case study molecules from the LocalRetro paper— where existing and strong template-based methods are successful —LLM-Syn-Planner successfully identified literature-supported routes. We show the routes here: https://drive.google.com/file/d/13S-G76_ZFzOH4l-R_WDysoGGcD4XVx0Z/view?usp=sharing. We verify each reaction with SciFinder and list the CAS reference. For reactions without literature precedent, we provide a plausible explanation and note potential reactivity problems for transparency. We emphasize that the same template limitations are present for existing template-based methods.
> > >
> > > Importantly, LLM-Syn-Planner uses the same reaction template database and stock (eMolecules) as prior methods, so it faces the same limitations. Should we have access to better templates such as from SYNTHIA [4], our framework naturally benefits from this.
> > >
> > > [1] "Deep retrosynthetic reaction prediction using local reactivity and global attention." JACS Au 1.10 (2021)
> > >
> > > [2] Retro*: Learning Retrosynthetic Planning with Neural Guided A* Search, ICML2020
> > >
> > > [3] "Re-evaluating retrosynthesis algorithms with syntheseus." Faraday Discussions 256 (2025)
> > >
> > > [4] https://www.synthiaonline.com/
> > >
> > > **Update**
> > >
> > > We thank the reviewer for their feedback and positive endorsement of our work! We are happy to answer any other questions that may arise and are very appreciative of their time and effort during this rebuttal period.

---

### Official Review · Reviewer_kcQx · 2025-03-13

**Overall Recommendation:** 3

**Summary:**

This work attempts to investigate whether LLMs can solve complex chemical sequences and proposes a framework that uses LLMs without tuning for conducting retrosynthesis and molecular design tasks. An effective method for formatting synthesis sequences is introduced. Additionally, a novel approach for searching sequence-level decisions via a reward function and partial feedback is presented, enabling multi-step retrosynthesis. Through this unified framework, the authors demonstrate the efficacy of the proposed method for chemical synthesis tasks.

**Claims And Evidence:**

Rather than claiming that LLMs can outperform or surpass previous specialized approaches, the authors investigate the capability of LLMs in chemical domains by assessing their problem-solving ability in complex tasks. Without relying on tuning methods such as fine-tuning or instruction tuning, they solely depend on parameterized and retrieved knowledge, as well as the models' own reasoning abilities. They design a well-structured framework that combines prompts and the GA algorithm, targeting reaction pathway optimization.

**Essential References Not Discussed:**

This work effectively covers the key references.

**Experimental Designs Or Analyses:**

Comparisons with other specialized models and experiments on two tasks are well conducted. However, experiments using other general LLMs are necessary to genuinely investigate the capability of LLMs in solving complex problems in the chemistry domain.

**Methods And Evaluation Criteria:**

The proposed framework utilizes a reaction database, LLM, and the GA algorithm with partial feedback and rewards. As a unified scheme, it presents a reasonable combination and demonstrates effective performance. The evaluation criteria are well-established for assessing the proposed framework.

**Other Comments Or Suggestions:**

This framework uses only GPT-4o, so experiments with other general LLMs are necessary. A cost analysis is also needed, as the 30-minute call limit per molecule could be expensive.

**Other Strengths And Weaknesses:**

It is a clever combination of existing modules, enhanced with curated feedback and a reward function. In essence, the proposed scheme is an extension of MOLLEO into the retrosynthesis task, with added reward and feedback. Solving multi-step retrosynthesis and molecular design using LLMs without tuning seems reasonable, but the technical novelty appears somewhat limited.

**Questions For Authors:**

Looking at the prompt template, it contains many explanations and instructions. Is the proposed scheme robust to variations in the prompt format?

**Relation To Broader Scientific Literature:**

This paper is related to machine learning-based organic retrosynthesis and molecular optimization works. It could be further extended to an autonomous laboratory framework or a molecular synthesis and generation pipeline using LLMs.

**Theoretical Claims:**

There are no theoretical claims.

---

> ### Author Rebuttal · Authors · 2025-04-01
>
> We thank the reviewer for their positive feedback! We are thrilled the reviewer finds our LLM-based retrosynthesis framework “could be further extended to an autonomous laboratory framework or a molecular synthesis and generation pipeline using LLMs”.
>
> Below, we address the reviewer’s questions about experiments as well as the technical contribution of the paper. We have provided additional experiment results in the following link: https://drive.google.com/file/d/1ANfpzJLJgJh83saOwBzK4AMgenb41Jnc/view?usp=sharing
>
> >Performance of LLM-Syn-Planner
>
> First, we conducted error analysis for LLM-Syn-Planner and observed that the main shortcomings stem not from the LLM itself but from the partial reward mechanism—namely, that a strong SCScore does not guarantee a molecule is in the purchasable set. Consequently, the algorithm can get stuck in an infinite loop.
>
> To address this, we simply added a penalty for repeated entries, which substantially improved performance and, as shown in Table 1, allowed LLM-Syn-Planner to outperform baselines.
>
> >Other general LLMs
>
> We have included new experimental results for the open-source models DeepSeek V3 and DeepSeek R1. The performance of DeepSeek V3 is shown in Table 1, while the results for DeepSeek R1 appear in Table 4. Interestingly, although DeepSeek R1 is designed as a reasoning model, it performs worse than DeepSeek V3. Moreover, because DeepSeek R1 includes its thinking process in the output, its overall cost is roughly three times higher than that of DeepSeek V3.
>
> >Technical contribution
>
> Our primary goal in this work is to investigate how effectively LLMs can be leveraged for retrosynthesis tasks. The key contribution of this work is to apply the idea of an evolutionary algorithm to generate the entire decision trajectory to the problem of retrosynthesis (which is new). In addition, the workflow we propose such as how LLMs can serve as the proposal and how a proper proxy and retrieval can augment the process is new. We will rephrase the text to reflect this. The strongest contribution, in our humble opinions, of this work is the demonstration of the strong performance of the retrosynthesis planning problem with our LLM-based tree-sampler EA method. To the best of our knowledge, we are the first to show that LLMs can achieve comparable or even better performance compared with specialized models. This also provides evidence for solving other science problems with a similar structure.
>
>
> >A cost analysis is also needed
>
> In Figure 1, we present a cost analysis showing the average cost of each discovered route across the four datasets. As the route length increases, the cost of finding that route also rises, but at an acceptable rate. Given the rapid advancements in LLMs, we anticipate that these costs will decrease over time. For example, Llama 3 outperforms Llama 2 significantly despite being released only one year later.
>
> >Ablation of the prompt
>
> In Table 2, we present an ablation study examining how the <Explanation> component in the overall prompt and the ‘Rational’ tag at each step affect performance. Our findings show that both elements enhance the language model’s reasoning capabilities, underscoring the importance of including thinking/explanation steps in the prompt.
>
> We are happy to address any additional comments or concerns the reviewer may have. Otherwise, we would appreciate it if the reviewer continues to view this work positively. Thank you once again for your time and thoughtful feedback.

---

### Official Review · Reviewer_hjfV · 2025-03-14

**Overall Recommendation:** 3

**Summary:**

This paper explores the potential of LLMs in addressing retrosynthesis planning and synthesizable  molecular design tasks. The authors first formulate retrosynthesis planning as a sequential decision-making problem with sequential route format that is suitable for LLMs. The authors introduce an evolutionary search algorithm called LLM-Syn-Planner, which allows LLMs to generate and optimize entire retrosynthetic pathways directly. They evaluate the algorithm on multiple USPTO and Pistachio datasets.  The results show that LLM-Syn-Planner performs comparably to or better than these traditional single-step  methods, but not yet competitive as single-step predictors in MCTS/Retro*. For synthesizable molecular  design, LLM-Syn-Designer, an extension of LLM-Syn-Planner, effectively balances optimization efficiency and synthesizability.

**Claims And Evidence:**

Yes, the claims are supported by clear and convincing experimental evidence.

**Essential References Not Discussed:**

No, related works are properly discussed.

**Experimental Designs Or Analyses:**

Yes, the experimental designs appear sound, particularly in evaluating retrosynthesis planning and  synthesizable molecular design using datasets like USPTO and Pistachio.

**Methods And Evaluation Criteria:**

Yes, the proposed methods and evaluation criteria are well-suited for the problem studied in the paper.

**Other Comments Or Suggestions:**

* A clear and formal definition of the functions INITIALIZATION and MUTATION , including their arguments and return values, would improve clarity.
* line 228: should R_0 be P_0
* Line 231: t should start from 0

**Other Strengths And Weaknesses:**

Strengths:
* The paper is clearly written and well-formatted.
* Unlike previous methods that use LLMs as single-step reaction predictors, this work introduces a novel  evolutionary approach that allows LLMs to generate, evaluate, and modify pathways at the route level.
* The proposed algorithm can be easily adapted to molecular design problems, unifying LLM-augmented
reaction decision programs.
* The code is available, enhancing reproducibility.

Weaknesses:
* The experiments are limited to GPT-4o as the only LLM tested. The authors could expand their evaluation to include other models, such as open-source models like Qwen-2.5, Llama-3, or recent strong reasoning models like o1 or r1.
* The evaluation metrics are limited to success rate. Including additional metrics such as route lengths and costs would provide a more comprehensive assessment of the proposed methods.

**Questions For Authors:**

* Does the solve rate continue to improve with more model calls (N)?
* For reaction-level evaluation, how is the chemical feasibility of the proposed reaction assessed?
* For the LLM as a single-step prediction model, how does the LLM identify substructures, and in what form are these substructures represented and used to extract templates? (line 197)

**Relation To Broader Scientific Literature:**

The key contributions of this paper are closely related to the broader field of AI for Science, particularly in chemistry and drug discovery. This aligns with the growing trend of using AI, including LLMs, to tackle complex scientific problems, as explored in works like [Segler'18] and [Chervonyi'25].

[Segler'18] Segler, Marwin HS, Mike Preuss, and Mark P. Waller. "Planning chemical syntheses with deep
neural networks and symbolic AI." Nature 555.7698 (2018): 604-610.

[Chervonyi'25] Chervonyi, Yuri, et al. "Gold-medalist Performance in Solving Olympiad Geometry with
AlphaGeometry2." arXiv preprint arXiv:2502.03544 (2025).

**Theoretical Claims:**

No, there are no proofs or theoretical claims in the paper.

---

> ### Author Rebuttal · Authors · 2025-04-01
>
> We thank the reviewer for their positive feedback! We are glad that the reviewer considers our method to be “novel” and our paper to be “well-written”. We wholeheartedly agree with the reviewer that this method can “be easily adapted to molecular design problems, unifying LLM-augmented reaction decision programs”.
>
> We have uploaded the results of the new experiments and the required algorithm box in https://drive.google.com/file/d/1IsuWkKzzLrNM0aYdbKx-3n2YnqkaECgg/view?usp=sharing
>
> >New LLM experiments:
>
> We first did an error analysis for LLM-Syn-Planner and observed that the main errors in LLM-Syn-Planner mainly come from the partial reward mechanism—a strong SC Score does not guarantee a molecule is in the purchasable set. Consequently, the algorithm can get stuck in an infinite loop. To address this, we simply added a penalty for repeated entries, which substantially improved performance and, as shown in Table 1, allowed LLM-Syn-Planner to outperform baselines.
>
> Based on this, we conducted experiments on an open-source model DeepSeek-V3, and the results are shown in Table 1. Both LLMs achieve comparable or even surpass the baseline specialized models, which demonstrates the effectiveness of the LLM-Syn-Planner framework. To the best of our knowledge, we are the first to show that current LLMs can achieve decent performance on multi-step retrosynthesis planning tasks, paving the way for an autonomous LLM agent framework for molecule synthesis and optimization.
>
> Moreover, we also tested DeepSeek R1. The results are shown in Table 4. Interestingly, although DeepSeek R1 is designed as a reasoning model, it performs worse than DeepSeek V3.
>
> >Evaluation metrics
>
> We conducted a cost analysis shown in Figure 1. The figure shows the average cost of each discovered route across the four datasets. As the route length increases, the cost of finding that route also rises, but at an acceptable rate. As open-source LLMs improve, it is reasonable to believe that LLM-Syn-Planner will also improve, so computational cost will be mitigated in the near future. For the route length, due to the time constraint for rebuttal, we will add it in our future version.
>
> > Definition of the functions INITIALIZATION and MUTATION
>
> We show the definition in Algorithm 1 and Algorithm 2.
>
> INITIALIZATION queries the LLM to produce an initial route for the given target molecule. It uses a RAG strategy by first identifying molecules similar to target_molecule in retrieval_db, retrieving their known routes, and providing these references to the LLM to guide route generation.
>
> MUTATION prompts the LLM to revise any invalid parts of current_route beginning at invalid_step_index. The LLM may directly fix the incorrect step, substitute it with a more suitable reaction, or even propose an alternative multi-step path if needed.
>
> >typo in the algorithm box
>
> Yes in line 228, R_0 should be P_0. In line 231, t should start from 0. Thanks for pointing out. We will fix this typo in the next version.
>
> >solve rate with more model calls (N)?
>
> When the number of model calls for LLM-Syn-Planner increases from 100 to 300, the solve rate shows a significant improvement. However, increasing the number of model calls from 300 to 500 yields only minimal additional gains. One factor is that our partial reward does not always correlate with final synthetic success (e.g., a molecule with an SC score of 1 might still be extremely difficult to synthesize in practice).
>
> >For reaction-level evaluation, how is the chemical feasibility of the proposed reaction assessed?
>
> To assess the chemical feasibility at the reaction level, we verify whether the product molecule in this step could validly undergo the backward reaction. In practice, we use the rdchiralRun function from the rdchiral package to check if the product matches the reaction template such that the backward reaction is recognized. If rdchiralRun confirms that the product can serve as a valid reactant under the backward reaction, we regard the forward reaction as chemically feasible.
>
> > How does the LLM identify substructures, and in what form are these substructures represented and used to extract templates? (line 197)
>
> We begin by prompting the LLM to identify relevant substructures and functional groups in the product molecule. In response, the LLM provides these substructures in SMILES format. Next, we verify each suggested substructure SMILES to ensure it is indeed present within the product structure. Once validated, we use these SMILES to compute Tanimoto similarities between the identified substructures and the product molecules in our reference reaction database to extract templates.
>
> We hope that our responses were sufficient in clarifying all the great questions asked by the reviewer. We sincerely thank the reviewer for their time and consideration. We would be happy to engage in further discussions if the reviewer has any additional questions or concerns.

---

### Official Review · Reviewer_diPA · 2025-03-20

**Overall Recommendation:** 3

**Summary:**

This paper explores the use of Large Language Models (LLMs) for the challenging task of multi-step retrosynthesis planning and extends this to synthesizable molecular design. The authors introduce a novel approach, LLM-Syn-Planner, which uses an evolutionary search algorithm where the LLM generates and optimizes entire retrosynthetic pathways. The method employs a sequential decision-making representation for synthesis routes, a three-level evaluation process (molecule, reaction, route), and a partial reward mechanism based on the SC score for incomplete routes. Experiments are conducted on USPTO and Pistachio datasets, comparing the LLM-augmented approach with traditional single-step models combined with search algorithms.

**Claims And Evidence:**

Claim: "We propose a novel way of searching sequence-level decisions with a smooth reward function and partial feedback by sampling from the space of decision sequences (full multi-step synthetic pathways) rather than individual states (a single reaction step)."

1. "Novel way of searching sequence-level decisions..."

Evidence: The paper does describe a search process operating at the level of entire synthesis routes (sequences of decisions). This contrasts with many traditional retrosynthesis methods that use search algorithms (like MCTS or A*) combined with single-step prediction models. The LLM-Syn-Planner generates entire pathways, which are then evaluated and mutated.

Problematic Aspects: The novelty here is questionable, and this is where the claim is weakest. Evolutionary algorithms that operate on entire sequences are not new.  Using an LLM to generate those sequences within an evolutionary framework is a new application, but the underlying search mechanism itself is not novel.  The paper needs to be very careful about how it frames this as "novel." It's a novel application of an evolutionary algorithm, but not a novel search algorithm itself.

2.  "...with a smooth reward function and partial feedback..."

  Evidence: The paper does describe a reward function that incorporates the `sc_score` (synthetic complexity score) of intermediate molecules. This is presented as a form of "partial reward" because it provides feedback on incomplete routes, unlike a binary reward that only rewards complete, valid syntheses. The `sc_score` itself acts as a smoother measure of progress than a simple binary success/failure.

Problematic Aspects: The use of intermediate rewards in reinforcement learning and search is very common; it's not inherently novel. The paper needs to acknowledge this and position its contribution appropriately. The partial feedback by comparing it against one with only the final reward. It is also very common.

**Essential References Not Discussed:**

Nope

**Experimental Designs Or Analyses:**

I have carefully examined the experimental designs and analyses presented in the paper, focusing on the comparison to baselines (Table 2) and the ablation studies (Tables 4, 5, and 6). Several issues undermine the soundness and validity of the experimental results:

Underperformance:
The most significant concern is the underperformance of LLM-Syn-Planner compared to several non-LLM baselines, as shown in Table 2.

Limited Ablation Studies:
While the ablation studies are a positive step, they are limited in scope. The authors should consider ablating other key components of the method, such as the partial reward mechanism, and perform a more thorough sensitivity analysis of the evolutionary algorithm's hyperparameters. The ablation results also suffer from the same lack of statistical significance.

**Methods And Evaluation Criteria:**

The paper's proposed methods and evaluation criteria are generally appropriate for the task of retrosynthesis planning and synthesizable molecular design. The use of an evolutionary algorithm with an LLM as a mutation operator is a reasonable approach, and the design choices, such as the sequential route representation, partial reward based on sc_score, and retrieval-augmented generation, are well-justified. The evaluation on standard benchmark datasets (USPTO, Pistachio) using common metrics (solve rate) allows for comparison with prior work. However, concerns remain regarding the potential computational cost of the method and the need for reporting statistical significance and more detailed analysis for molecular design. Furthermore, the chosen experimental methodology, while sound, resulted in performance that did not surpass existing, non-LLM approaches. This indicates that, while the methods make sense, they do not demonstrate an advantage in the current implementation.

**Other Comments Or Suggestions:**

Nope

**Other Strengths And Weaknesses:**

Strengths:
1. Timely and Relevant Topic. The use of LLMs in scientific domains, particularly for complex decision-making, is a highly relevant and active area of research. Applying LLMs to retrosynthesis, a core challenge in organic chemistry, is a timely and appropriate topic for ICML.

2. Exploration of Route-Level Generation: The paper's attempt to move beyond single-step prediction and generate entire synthetic routes is a notable strength. This approach aims to leverage LLMs' potential for long-range planning, which is crucial for tackling the combinatorial complexity of retrosynthesis.

3. Innovative Combination of Techniques:  The integration of an evolutionary algorithm with LLM-based mutation and RAG is a well-motivated approach. The use of partial rewards, sequential representation, and RAG addresses some known limitations of LLMs in structured output generation and long-horizon planning tasks.

4.  Evaluation on Multiple Tasks: Evaluating the method on both retrosynthesis planning and synthesizable molecular design demonstrates a broader applicability and strengthens the empirical assessment.

Weaknesses:
1. Limited Novelty: While the combination of techniques is specific to this work, the individual components (evolutionary algorithms, RAG, LLM prompting for chemistry) are well-established. The paper's core novelty lies in the specific application and integration, which might be considered incremental by some reviewers.

2. Underperformance Compared to Non-LLM Baselines: The most significant weakness is the empirical performance.  Table 2 clearly shows that LLM-Syn-Planner, while performing well in absolute terms, underperforms compared to several established non-LLM baselines (Retro*, and MCTS combined with specialized single-step models) on the standard retrosynthesis planning tasks. This significantly weakens the claim of the method's effectiveness in practice.  The paper would be significantly stronger if it could demonstrate a clear advantage over existing methods.

3. Lack of Insight into LLM Reasoning: The paper doesn't deeply analyze why the LLM generates the routes it does.  Is it truly leveraging chemical knowledge, or is it relying on pattern matching from the training data? Understanding the LLM's internal reasoning (or lack thereof) would provide valuable insights. More error analysis or qualitative examples of successful/failed routes would be beneficial.

4. Scalability Concerns (Potentially): While not explicitly addressed, the computational cost of using an LLM within an evolutionary loop might be significant.  The paper mentions a 30-minute time limit per molecule, but further discussion of scalability and efficiency would be helpful.

**Questions For Authors:**

Line 91, "At each step, we use a backward reaction to decompose", can authors clarify how this backward reaction is implemented?

**Relation To Broader Scientific Literature:**

The key contributions of the paper should be carefully positioned within the existing scientific literature. While the application of LLMs to retrosynthesis is a timely and relevant topic, several aspects of the proposed method are not fundamentally novel. Specifically:

The use of an evolutionary algorithm for search is well-established in chemistry, particularly for molecular design (e.g., Jensen, Jan H. "A graph-based genetic algorithm and generative model/Monte Carlo tree search for the exploration of chemical space." Chemical science 10.12 (2019): 3567-3572.). The authors should acknowledge prior work in this area and clearly differentiate their contribution.

The concept of operating on entire synthesis routes (sequences of decisions) is not unique to this work. While many retrosynthesis methods focus on single-step prediction, others, such as planning algorithms and prior uses of evolutionary algorithms, also consider longer-range dependencies.

The use of an LLM as a genetic operator is most closely related to the MolLEO paper (Wang et al., 2024), which applies a similar idea to molecular optimization. The authors should explicitly discuss this connection and highlight the differences in their approach.

Partial rewards and retrieval-augmented generation (RAG) are general techniques, not specific to this work. The authors should acknowledge their broader use in reinforcement learning and NLP.

The paper's primary contribution lies in the specific combination of these techniques within an LLM-driven framework for retrosynthesis. The authors should emphasize the application and integration of these ideas, rather than claiming broad novelty for the individual components. A more thorough discussion of related work, clearly positioning the paper within the existing literature, is crucial.

**Theoretical Claims:**

Nope

---

> ### Author Rebuttal · Authors · 2025-04-01
>
> We thank the reviewer for their feedback and thoughtful questions. We are ecstatic that the reviewer finds our method “crucial for tackling the combinatorial complexity of retrosynthesis”. We have updated the new results here: https://drive.google.com/file/d/1Car72zPjMG9__41PfJAGbMkbNObYHjBS/view?usp=sharing
>
> >Error analysis
>
> We first did an error analysis and observed that the main errors in LLM-Syn-Planner mainly come from the partial reward mechanism—a strong SC Score does not guarantee a molecule is in the purchasable set. Consequently, the algorithm can get stuck in an infinite loop. To address this, we simply added a penalty for repeated entries, which substantially improved performance and, as shown in Table 1, allowed LLM-Syn-Planner to outperform baselines.
>
> >Underperformance
>
> After resolving the infinite loop issue in our error analysis, we present the updated results in Table 1. LLM-Syn-Planner now achieves superior performance compared to the baseline specialized models. Additionally, we include a new open-source LLM, DeepSeek-V3. Our findings show that both LLMs perform well within our framework, demonstrating the effectiveness of LLM-Syn-Former.
>
> > Additional ablation studies
>
> We conducted several ablation studies to evaluate different design choices: route format, the use of molecule RAG, the reward signal, EA parameters, and prompt robustness.
>
> In a separate ablation study for direct user queries (Table 3), we queried each LLM 100 times for a fair comparison. On the USPTO easy dataset, both LLMs solved fewer than 10 routes, and on the Pistachio Hard dataset, they failed in nearly all cases. These results suggest the models do not merely “remember” synthetic routes from their training data.
>
> Additionally, Figure 1 provides a cost analysis, while Table 4 presents an ablation study using DeepSeek-R1, and Table 5 studies the statistical significance.
>
>
> >problematic claim & our technical contribution
>
> We agree with the reviewer the search algorithm itself is not new. The key contribution of this work is to apply the idea of an EA to generate the entire decision trajectory to the problem of retrosynthesis (which is new). In addition, the workflow we propose such as how LLMs can serve as the proposal and how a proper proxy and retrieval can augment the process is new. We will rephrase the text to reflect this.
>
> Our primary goal in this work is to investigate how effectively LLMs can be leveraged for retrosynthesis tasks. The main contribution is our demonstration that an LLM-based tree-sampler EA method can deliver strong performance in multi-step retrosynthesis planning—surpassing even specialized models. We believe this finding not only validates the utility of LLMs in retrosynthesis but also offers a valuable contribution to the broader research community.
>
>
> >acknowledge EA for molecule design, Partial reward in RL and RAG in NLP
>
> Thank you for pointing out the literature. We cited Jensen 2019 and here we adopt a similar EA framework, but we design it for a sequential decision-making problem that is highly different (search for a single molecule vs a synthesis route). We use LLMs as the proposal with additional consideration of partial reward and retrieval-based augmentation. We will acknowledge them in the next version.
>
> >Compared with MolLEO
>
> Our work and MolLEO share only the concept of combining EA with LLMs, but address fundamentally different domains. For task domains, MolLEO focuses on small molecule optimization (naturally represented in SMILES strings). But LLM-Syn-Planner is designed for a sequential decision-making problem. For task challenges, molecule optimization uses LLM to directly edit the molecule and can easily get ground truth feedback from an oracle function. However, retrosynthesis planning poses a more complex, sequential decision-making challenge within a constrained search space.
>
> >Scalability Concerns
>
> We have shown that the open-source model DeepSeek-V3 can outperform specialized models, suggesting that as open-source LLMs continue to improve, so will LLM-Syn-Planner, thereby reducing computational costs. Currently, the primary efficiency bottleneck is the OpenAI API rate limit, which caps the number of tokens processed each minute. Another bottleneck stems from the raw reaction database; classifying or indexing these reactions would improve the efficiency of RAG.
>
> >question about backward reaction
>
> In retrosynthesis planning, especially in top-down planning algorithms, the reaction is considered in reverse. We denote this by writing the reaction as Product >> Reactant1.Reactant2, is often referred to as a backward reaction.
>
> We sincerely thank the reviewer for their valuable feedback. We hope our rebuttal has effectively addressed their questions and concerns. If the reviewer is satisfied with our responses, we kindly ask them to consider a reassessment of our paper. We remain more than happy to address any additional questions or concerns that may arise.

---

### Decision · Program_Chairs · 2025-05-01

**Decision:**

Accept (poster)

**Comment:**

This paper introduces a framework for LLM-driven multi-step retrosynthesis and synthesizable molecular design. The main contributions include:
- A novel integration of evolutionary algorithms with LLMs for sequence-level decision-making in retrosynthesis, enabling end-to-end pathway generation and refinement.
- A reward mechanism incorporating partial feedback (e.g., SCScore) for intermediate route assessment, and the use of retrieval-augmented generation to improve chemical validity.
- Empirical evaluation across multiple retrosynthesis and molecular design benchmarks (USPTO, Pistachio), with evidence showing that LLM-Syn-Planner matches or outperforms specialized baselines in several settings, even without fine-tuning.

The authors provided extensive responses, including new experiments with open-source LLMs (DeepSeek), additional ablation studies, error analysis, and clarifications of reaction verification and deployment cost. Reviewer diPA upgraded their score after acknowledging improved performance. Reviewer cmv4 remains concerned about practical cost, but noted improvements and raised his score.

Despite some lingering concerns, the reviewers unanimously support its acceptance.

The authors are suggested to (1) clarify **novelty claims to focus on the integration and application, not algorithmic development**, (2) address cost-effectiveness and fairness concerns more transparently, and (3) deepen discussion on real-world generalizability, including LLM limitations and verification of synthetic validity.